# Investigating the Overlooked Hessian Structure: From CNNs to LLMs

Qian-Yuan Tang* [1]   Yufei Gu* [2]   Yunfeng Cai [3]   Mingming Sun [4]   Ping Li [5]   Xun Zhou [6]   Zeke Xie† [2]

## Abstract

It is well-known that the Hessian of deep loss landscape matters to optimization and generalization of deep learning. Previous studies reported a rough Hessian structure in deep learning, which consists of two components, a small number of large eigenvalues and a large number of nearly-zero eigenvalues. To the best of our knowledge, we are the first to report that a simple but overlooked power-law Hessian structure exists in well-trained deep neural networks, including Convolutional Neural Networks (CNNs) and Large Language Models (LLMs). Moreover, we provide a maximum-entropy theoretical interpretation for the power-law Hessian structure and theoretically demonstrate the existence of a robust and low-dimensional subspace of deep neural networks. Our extensive experiments using the proposed power-law spectral method demonstrate that the power-law Hessian spectra critically relate to multiple important behaviors of deep learning, including optimization, generalization, and overparameterization. Notably, we discover that the power-law Hessian structure of a given LLM can often predict generalization during training in some occasions, while conventional sharpness-based generalization measures which often work well on CNNs largely fail as an effective generalization predictor of LLMs.

## 1. Introduction

It is well-known that the Hessian matters to optimization, generalization, and even robustness of deep learning (Li et al., 2020; Ghorbani et al., 2019; Jacot et al., 2019;

[1]Department of Physics, Hong Kong Baptist University [2]xLeaF Lab, The Hong Kong University of Science and Technology (Guangzhou) [3]BIMSA [4]AGI Lab, BIMSA [5]Rutgers University [6]Seed-Foundation-Model Team, ByteDance. ∗: Equal Contributions. Correspondence to: † Zeke Xie <zekexie@hkust-gz.edu.cn>.

*Proceedings of the 42nd International Conference on Machine Learning*, Vancouver, Canada. PMLR 267, 2025. Copyright 2025 by the author(s).

Yao et al., 2018; Dauphin et al., 2014; Byrd et al., 2011). Deep learning usually finds flat minima that generalize well (Hochreiter & Schmidhuber, 1995; 1997). The Hessian is one of the most important measures of the minima flatness and directly relates to generalization in deep learning (Hoffer et al., 2017; Neyshabur et al., 2017; Dinh et al., 2017; Wu et al., 2017). Jiang et al. (2019) reported that minima-flatness-based generalization bound is still the most reliable metric in extensive experiments. Wu et al. (2017) reported that the low-complexity solutions that generalize well have a small norm of Hessian matrix with respect to model parameters. Yao et al. (2018) reported that the spectrum of the Hessian closely connects to large-batch training and adversarial robustness.

A number of works empirically studied the Hessian structure in Deep Neural Networks (DNNs). Some papers (Sagun et al., 2016; 2017; Wu et al., 2017) empirically reported a two-component structure that, in the context of deep learning, most Hessian eigenvalues are nearly zero, while a small number of eigenvalues are large. Sankar et al. (2021) revealed that the layer-wise Hessian spectrum is similar to the entire Hessian spectrum. Zhang et al. (2024b) demonstrated that SGD performs worse than Adam for Transformers when Hessian spectra exhibit blockwise heterogeneity. Ormaniec et al. (2024) theoretically studied one-layer Transformers' Hessian in matrix derivatives while comparing them to classical networks in deep learning.

However, quantitative or statistical analysis of the Hessian structure is still largely under-explored for modern neural networks. Does an elegant statistical structure hide behind the Hessian spectrum? Does such structure matter to CNNs and LLMs? Our work provides a novel approach to understanding and analyzing the Hessian structure of deep loss landscape from a statistical perspective. This work mainly made three contributions.

First, to the best of our knowledge, we are the first to empirically discover and statistically test the power-law Hessian structure of deep loss landscape which has been overlooked by previous studies. Such novel power-law structure widely exist in DNNs, including CNNs and LLMs.

Second, we propose a framework of power-law spectral analysis for quantitatively analyzing the Hessian structure in deep learning. We not only reveal how the power-law

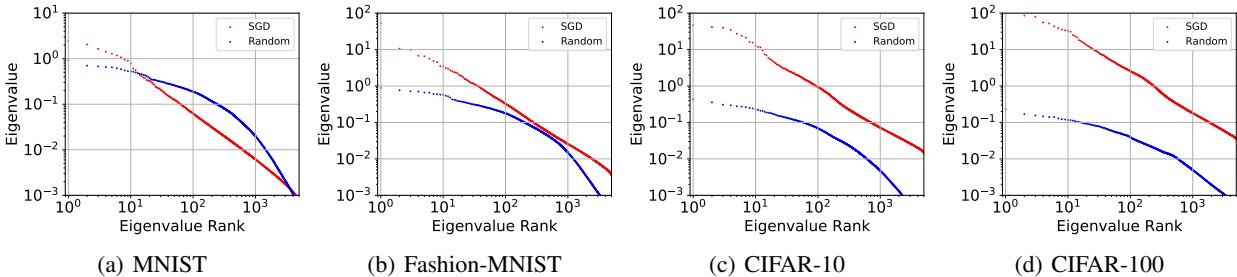

(a) MNIST  (b) Fashion-MNIST  (c) CIFAR-10  (d) CIFAR-100

*Figure 1.* The power-law structure of the Hessian spectrum in deep learning. Model: LeNet. We may clearly observe that the power-law spectra generally hold for well-trained deep networks on various natural or artificial datasets, while do not hold for random neural networks. We also report that a small number of outlier eigenvalues ($\sim 10$) slightly deviate from the fitted straight line.

spectra explain the theoretical origin of striking findings but also empirically demonstrate multiple novel insights on optimization, generalization, and scaling.

Third, we report that the power-law Hessian spectral analysis can sometimes predict generalization of LLMs during training, particularly when conventional sharpness-based generalization measures that often work well on CNNs become nearly useless as a generalization predictor of LLMs. This suggests that generalization measures for LLMs remain to be deeply explored from a loss landscape perspective.

## 2. The Overlooked Hessian Structure

In this section, we demonstrate that the Hessian spectra of well-trained deep neural networks have a simple power-law structure overlooked by previous studies and how to theoretically derive the power-law structure.

**Notations.** We denote the training dataset as $\{(x, y)\} = \{(x_j, y_j)\}_{j=1}^N$ drawn from the data distribution $\mathcal{S}$, the $n$ model parameters as $\theta$ and the loss function over one data sample $\{(x_j, y_j)\}$ as $l(\theta, (x_j, y_j))$. For simplicity, we further denote the training loss as $L(\theta) = \frac{1}{N} \sum_{j=1}^N l(\theta, (x_j, y_j))$ and denote its Hessian as $H$. We write the descending ordered eigenvalues of the Hessian $H$ as $\{\lambda_1, \lambda_2, \ldots, \lambda_n\}$ and denote the spectral density as $p(\lambda)$.

### 2.1. Visualizing the Power-Law Structure

Hessian has been studied as a measure of minima flatness Dinh et al. (2017); Xie et al. (2021b) and loss curvature Achille & Soatto (2019), while these works failed to reveal its elegant statistical structure. To better understand the distribution of the Hessian spectrum, we first visualize the Hessian spectrum of a well-trained neural network and a randomly initialized neural network by using the Lanczos algorithm (Meurant & Strakoš, 2006; Yao et al., 2020) to estimate the eigenvalues and spectral densities. In Figure 1, we display the top 6000 eigenvalues and their corresponding rank order. Both axes are $\log$-scale. And we surprisingly

*Table 1.* The Kolmogorov-Smirnov statistics of the Hessian spectra of LeNets on various datasets. The estimated power exponent $\hat{\beta}$ and slope magnitude $\hat{s}$ are also displayed.

| Dataset | Model | Training | $d_{\mathrm{ks}}$ | $d_{\mathrm{c}}$ | Power-Law |
|---|---|---|---|---|---|
| MNIST | LeNet | Random | 0.0796 | 0.0430 | No |
| MNIST | LeNet | SGD | 0.00900 | 0.0430 | Yes |
| Fashion-MNIST | LeNet | Random | 0.0971 | 0.0430 | No |
| Fashion-MNIST | LeNet | SGD | 0.0132 | 0.0430 | Yes |
| CIFAR-10 | LeNet | Random | 0.0663 | 0.0430 | No |
| CIFAR-10 | LeNet | SGD | 0.0279 | 0.0430 | Yes |
| CIFAR-100 | LeNet | Random | 0.0944 | 0.0430 | No |
| CIFAR-100 | LeNet | SGD | 0.0315 | 0.0430 | Yes |

discover an approximately straight line fits the Hessian spectrum of the well-trained neural network surprisingly well, except that a small number of outliers ($\sim 10$) slightly deviate from the fitted straight line. To the best of our knowledge, these fitted power-law Hessian spectra were not empirically discovered or theoretically discussed by previous papers for neural networks in deep learning.

The well-fitted straight line means that the observed distribution of the Hessian eigenvalues of trained neural networks approximately obeys a power-law distribution,

$$p(\lambda) = Z_c^{-1} \lambda^{-\beta}, \qquad (1)$$

where $Z_c$ is the normalization factor. The observed eigenvalues can be considered as $n$ samples from the power-law distribution $p(\lambda)$. We may also use a corresponding finite-sample power law for describing the observed law as

$$f_k = \frac{\lambda_k}{\mathrm{Tr}(H)} = Z_d^{-1} k^{-\frac{1}{\beta-1}}, \qquad (2)$$

where $f$ is the trace-normalized eigenvalue, $k$ is the rank order, the trace $\mathrm{Tr}(H) = \sum_{k=1}^n \lambda_k$, and $Z_d = \sum_{k=1}^n k^{-\frac{1}{\beta-1}}$ is the normalization factor for the finite-sample power law. Note that the finite-sample power law is also called Zipf's law. This can also be approximately written as

$$\lambda_k = \lambda_1 k^{-s}, \qquad (3)$$

if we let $s = \frac{1}{\beta - 1}$ denote the power exponent of Zipf's law.

## 2.2. A Maximum-Entropy Interpretation

In this subsection, we show that the maximum entropy principle widely used in statistical physics may informally explain the power-law Hessian structure in deep learning.

The maximum entropy principle (Guiasu & Shenitzer, 1985), also named the maximum entropy prior, states that the probability distribution which best represents the current state of knowledge about a system at equilibrium is the one with the highest entropy. This principle indicates that if we have no prior knowledge for suspecting one state over any other, then all states can be considered equally likely for a system at equilibrium. The logarithmic space volume is often regarded as a kind of entropy in statistical physics (Visser, 2013). Note that flat minima have larger space volume reflected by $\det(H^{-1})$. It means maximizing the minima space volume for better generalization may be regarded as a kind of entropy maximization principle. Following Visser (2013), we may explicitly write the volume entropy as

$$S_{\text{vol}} = \log \det(H^{-1}) = -\int p(\lambda) \log \lambda d\lambda \qquad (4)$$

and the spectral entropy as

$$S_{\text{p}} = -\int p(\lambda) \log p(\lambda) d\lambda, \qquad (5)$$

which is the entropy of the spectral density distribution.

**Theorem 1** (The Maximum-Entropy Interpretation). *Suppose we have the volume entropy $S_{\text{vol}}$ as Equation (4) and the spectral entropy $S_{\text{p}}$ as Equation (5). To find the optimal distribution $p^{\star}(\lambda)$ that maximizes the total entropy, where $S_{\text{total}} = S_{\text{p}} + \beta_{\text{vol}} S_{\text{vol}}$ and $\beta_{\text{norm}}$ is a Lagrange multiplier, the optimal distribution $p^{\star}(\lambda)$ can be solved as*

$$p^{\star}(\lambda) = e^{-\beta_{\text{norm}}} \lambda^{-\beta_{\text{vol}}}. \qquad (6)$$

We leave the proof in Appendix A. We note that the result in Theorem 1 has an amazingly similar form to (1) with $\beta_{\text{norm}} = \log Z_c$ and $\beta = \beta_{\text{vol}}$.

We may interpret the power-law structure of the Hessian spectrum from two basic maximum entropy principles with the spectral density normalization constraint. It roughly means that simple rules can almost explain the power-law Hessian spectrum in deep learning as well as statistical physics (Visser, 2013). While previous work in the field of deep learning has also interpreted the minima flatness of neural networks from an entropy perspective (Baldassi et al., 2020), the spectra have much simpler structures as shown with our empirical results.

Interestingly, similar well-fitted power laws have been widely discussed in neuroscience and biology (Reuveni

et al., 2008; Tang & Kaneko, 2020). This exactly motivates us to further verify and study the power-law structure of the Hessian spectrum in the context of deep learning. We discover that the elegant power-law structure indeed exists in well-trained deep neural networks just like bioactive proteins. In contrast, random neural networks have no such power-law structure, just like deactivated (denatured or unfolded) proteins. Random neural networks which have no functional ability on the given task, break the power-law spectra similarly to deactivated proteins.

## 2.3. Goodness-of-fit Test

In this subsection, we are the first to conduct formal statistical tests on the Hessian structure. We also note that a recent work (Xie et al., 2023a) follow our statistical spectral analysis via KS Tests and only studied the structure of stochastic gradients rather than the Hessian structure.

Following Alstott et al. (2014), we use Maximum Likelihood Estimation (MLE) for estimating the parameter $\beta$ of the fitted power-law distributions and the Kolmogorov-Smirnov Test (KS Test) (Massey Jr, 1951; Goldstein et al., 2004) for statistically testing the goodness of the fit. The KS test statistic is the KS distance $d_{\text{ks}}$ between the hypothesized (fitted) distribution and the empirical data, which measures the goodness of fitting. Mathematically, the KS distance is defined as $d_{\text{ks}} = \sup_{\lambda} |F^{\star}(\lambda) - \hat{F}(\lambda)|$, where $F^{\star}(\lambda)$ is the hypothesized cumulative distribution function and $\hat{F}(\lambda)$ is the empirical cumulative distribution function based on the sampled data (Goldstein et al., 2004).

The estimated power exponent via MLE (Clauset et al., 2009) can be written as $\hat{\beta} = 1 + K \left[ \sum_{i=1}^{K} \ln \left( \frac{\lambda_i}{\lambda_{\text{cutoff}}} \right) \right]^{-1}$, where $K$ is the number of tested samples and we set $\lambda_{\text{cutoff}} = \lambda_k$. The Powerlaw library (Alstott et al., 2014) provides a convenient tool to compute the KS distance, $d_{\text{ks}}$, and estimate the power exponent.

According to the practice of KS Test, we first state **the power-law hypothesis** that the tested spectrum is power-law. If $d_{\text{ks}}$ is lower than the critical value $d_{\text{c}}$ at the $\alpha = 0.05$ significance level, the KS test statistically will support (not reject) the power-law hypothesis. The test results associated with Figure 1 are presented in Table 1. We leave the details and more test results (e.g., ResNet18) in Appendix F.

When we say that a spectrum is (approximately) power-law in this paper, we mean that the KS test provides positive evidence to the power-law hypothesis instead of rejecting the power-law hypothesis. Our KS test results reject the power-law hypothesis for random neural networks and do not reject the power-law hypothesis for well-trained neural networks. Moreover, when the power-law hypothesis holds, the KS distance is usually significantly smaller than the critical value $d_{\text{c}}$. For simplicity, the default $\alpha = 0.05$ significance

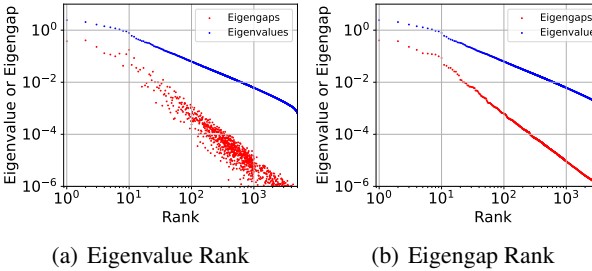

(a) Eigenvalue Rank    (b) Eigengap Rank

*Figure 2.* The power-law Hessian eigengaps. Model: LeNet. Datsets: MNIST. Subfigure (a) displayed the eigengaps by original rank indices sorted by eigenvalues. Subfigure (b) displayed the eigengaps by rank indices re-sorted by eigengaps. We also present the results of Fashion-MNIST in Figure 21 of Appendix D and GPT2-small in Figure 30 of Appendix E.

level is abbreviated in the following.

Following related work on the Hessian of neural networks (Thomas et al., 2020), our empirical analysis and statistical tests mainly focused on the top ($\sim 1000$) large eigenvalues larger than some minimal cutoff value $\lambda_{\text{cutoff}}$ for three reasons. First, focusing on relatively large values is very reasonable and common in various fields' power-law studies, as real-world distributions typically follow power laws only after/larger than some cutoff values (Clauset et al., 2009) to ensure the convergence of the probability distribution. Second, researchers are usually more interested in significantly large eigenvalues which contribute more to Hessian, minima sharpness, or generalization bound (Thomas et al., 2020). Third, empirically estimating a large number of nearly zero eigenvalues is very inaccurate and expensive.

### 2.4. The Power-Law Hessian Eigengaps

In this subsection, we report that the overlooked eigengaps of Hessian are also power-law and how the eigengaps suggest a robust and low-dimensional learning subspace.

The empirical investigation of the Hessian eigengaps is missing in previous works. Our experiments have closed this gap. Our experiments show that top eigengaps dominate other tailed eigengaps in deep learning. We visualize and verify the approximate power-law eigengaps in Figure 2.

The phenomenon of low-dimensional learning subspace was empirically reported recently (Gur-Ari et al., 2018; Ghorbani et al., 2019; Xie et al., 2021b) but still lacks theoretical understanding. Does the phenomenon theoretically depend on the Hessian structure? Our answer is yes. In the following part, we will demonstrate why the eigengaps of the Hessian $H$ may naturally lead to the phenomenon that learning dynamics mainly take place in a low-dimensional space during the entire training process.

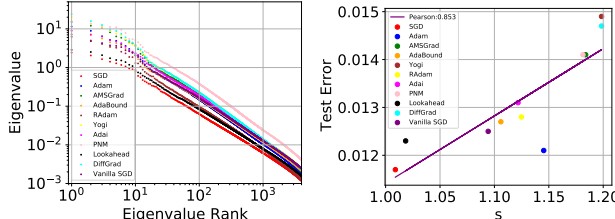

*Figure 3.* The power-law spectra hold across optimizers. Moreover, the slope magnitude $\hat{s}$ is an indicator of minima sharpness and a predictor of test performance. Model: LeNet. Dataset: MNIST.

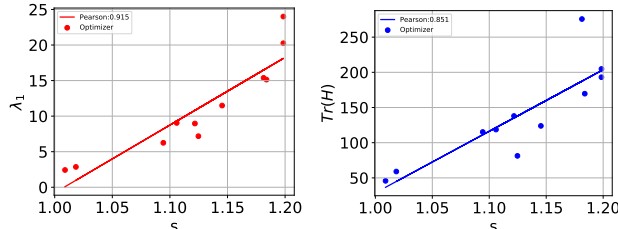

*Figure 4.* The slope magnitude $\hat{s}$ closely correlates to the largest Hessian eigenvalue and the Hessian trace. Model: LeNet. Dataset: MNIST.

To quantitatively understand why learning subspace is robust[1], we may use the angle between the original Hessian eigenvector $u_k$ and the perturbed Hessian eigenvector $\tilde{u}_k$, namely $\langle u_k, \tilde{u}_k \rangle$. Suppose the original Hessian is $H$, the perturbed Hessian is $\tilde{H} = H + \epsilon M$, the $i$-th eigenvector of $H$ is $u_i$, and its corresponding perturbed eigenvector is $\tilde{u}_i$. Under the conditions of the Davis-Kahan Theorem and (13), we have

$$\sup \sin\langle u_k, \tilde{u}_k \rangle = \frac{2\epsilon \|M\|_{op}}{\min(\lambda_{k-1} - \lambda_k, \lambda_k - \lambda_{k+1})}$$
$$= \frac{2\epsilon \|M\|_{op}(k+1)^{s+1}}{\lambda_1}, \qquad (7)$$

where $\|M\|_{op}$ is the operator norm of the perturbation $M$. In the derivation details, we applied Theorem 2, a useful variant of the Davis-Kahan Theorem (Yu et al., 2015), directly to the Hessian in deep learning and demonstrate that the eigenspace robustness (spanned by the eigenvectors) is relatively tight for the top-learning eigenspace. To the extent of our knowledge, we are the first to theoretically explain the robust and low-dimensional learning subspace using Hessian eigengaps. Formal theoretical analysis and more discussion are available in Appendix B.

## 3. Empirical Analysis of CNNs

In this section, we conduct extensive experiments for exploring the behaviors of deep learning through the lens of power-law spectral analysis.

---

[1]In this paper, robust space means that the space's dimensions are stable during training.

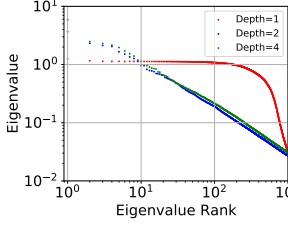
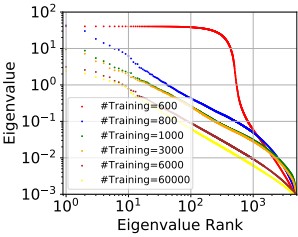

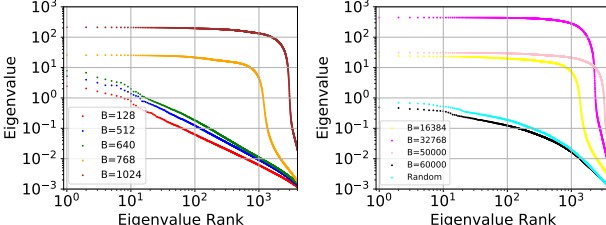

*Figure 5.* The power-law spectrum holds well in overparameterized deep models but disappears in the underparameterized single-layer FCN.

*Figure 6.* The spectra of LeNet on MNIST with respect to various numbers of training samples.

*Figure 7.* Batch size matters to the spectrum. We discover three phases of the Hessian spectra for large-batch training. Model: LeNet. Dataset: MNIST.

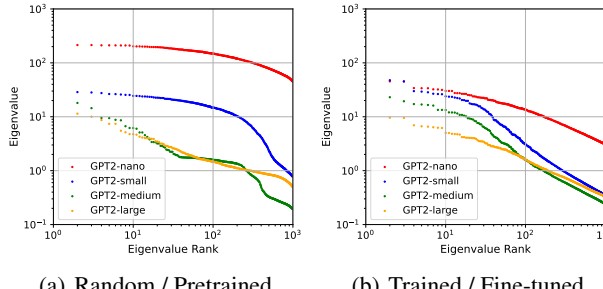

(a) Random / Pretrained    (b) Trained / Fine-tuned

*Figure 8.* The power-law Hessian structure exists in well-trained LLMs but disappear in their random initializations (GPT2-nano) or pretrained checkpoints (GPT2-small, GPT2-medium, GPT2-large). The power-law Hessian structure emerged on well-trained LLMs after training / fine-tuning.

**Models:** LeNet (LeCun et al., 1998), Fully Connected Networks (FCN), and ResNet18 (He et al., 2016).

**Datasets:** MNIST (LeCun, 1998), Fashion-MNIST (Xiao et al., 2017), CIFAR-10/100 (Krizhevsky & Hinton, 2009), and non-image Avila (De Stefano et al., 2018).

**1. Optimization and Generalization.** Figure 3 discovered that the power-law spectrum consistently holds for various popular optimizers, such as SGD, Vanilla SGD, Adam, AMSGrad, AdaBound, Yogi, RAdam, Adai, PNM, Lookahead, and DiffGrad, as long as the optimizers can train the network well. We present the KS test results in Table 3.

It is known that sharpness-based generalization measures are considered the most predictive generalization measures in deep learning (Jiang et al., 2019). We discover that the slope magnitude $\hat{s}$ of the fitted straight line may serve as a nice predictor of minima sharpness and generalization, when the power-law Hessian structure is well fitted. Note that it is common to measure minima's sharpness by the largest Hessian eigenvalue or the Hessian trace. A smaller $\hat{s}$ highly correlates to a smaller largest eigenvalue and a smaller trace in Figure 4. The similar observation holds on CIFAR-10 displayed in Figures 4 and 22 of Appendix D.

**2. Overparameterization.** Figure 5 shows that the power-law spectrum holds well in overparameterized models, but disappears in underparameterized models. Overparameterization is necessary for the power-law spectrum in deep learning. It will be interesting to study phase transition of under-parameterization to over-parameterization in future.

**3. The size of training data.** We evaluate the Hessian structure over various sized training data in Figure 6. The model trained with limited training data would break the power-law structure similarly as underparameterization, and lead to many sharp directions in the loss landscape. We see that data scaling and model scaling surprisingly exhibit very similar Hessian structures.

**4. Batch Size.** We discover the three different phases for large-batch training via the curves in Figure 7 and the KS test results in Table D of the appendix. To our knowledge, we are the first to report the phases and sharp phase transition for large-batch training. When we train CNNs with the same training epochs and let the batch size increase from 640 to 768, the power-law structure suddenly breaks. We see that inadequate training due to a large batch size exhibits a Hessian structure similar to one with limited training data. However, in Table D, training CNNs with the same iterations, we observe that large-batch training can also lead to power laws at the expense of more compute.

**5. Supplementary Results.** In Appendix D, we further discussed various interesting empirical results and insights, including studies on linear networks, modern architectures (such as ResNet18), noisy labels, task transferability, and the heavy-tail behavior of SGD.

## 4. Empirical Analysis of LLMs

In this section, we empirically studied how the power-law Hessian structure of LLMs behaves differently.

**Models:** GPT-2 family (Radford et al., 2019): GPT2-nano (11M), GPT2-small (124M), GPT2-medium (355M), and GPT2-large (774M), and TinyLlama (Zhang et al., 2024a) (1.1B-Chat-v1.0) with LoRA adapter (Hu et al., 2021).

*Table 2.* The Kolmogorov-Smirnov statistics of the Hessian spectra of various LLMs on various datasets.

| Dataset | Model | Training | $d_{ks}$ | $d_c$ | Power-Law |
|---|---|---|---|---|---|
| OpenWebText | GPT2-small | Random | 0.1168 | 0.0430 | No |
| OpenWebText | GPT2-small | Trained | 0.0418 | 0.0430 | Yes |
| Shakespeare | GPT2-nano | Random | 0.0969 | 0.0430 | No |
| Shakespeare | GPT2-nano | Fine-tuned | 0.0353 | 0.0430 | Yes |
| Shakespeare | GPT2-small | Pretrained | 0.1058 | 0.0430 | No |
| Shakespeare | GPT2-small | Fine-tuned | 0.0259 | 0.0430 | Yes |
| Shakespeare | GPT2-medium | Pretrained | 0.0787 | 0.0430 | No |
| Shakespeare | GPT2-medium | Fine-tuned | 0.0184 | 0.0430 | Yes |
| Shakespeare | GPT2-large | Pretrained | 0.0496 | 0.0430 | No |
| Shakespeare | GPT2-large | Fine-tuned | 0.0160 | 0.0430 | Yes |
| MathQA | Tinyllama (LoRA) | Random | 0.0552 | 0.0430 | No |
| MathQA | Tinyllama (LoRA) | Fine-tuned | 0.0249 | 0.0430 | Yes |

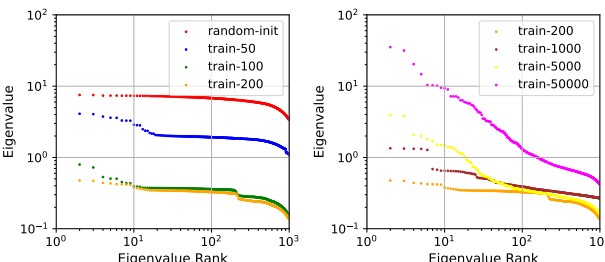

*Figure 9.* In the pretraining experiment of GPT2, the power-law Hessian structure emerged as training progressed in a two-stage process. In the first stage, the Hessian eigenvalues decrease in magnitude, indicating the discovery of a flat minimum. In the second stage, the primal Hessian eigenvalues increase to form a power-law distribution, reflecting a transition to a sharper minimum. Model: GPT2-small. Dataset: `OpenWebText`.

**Datasets:** OpenWebText (Gokaslan et al., 2019), Shakespeare (Karpathy, 2015), and MathQA (Amini et al., 2019).

**1. Pre-training and Fine-tuning of LLMs.** We empirically investigate the Hessian structure of LLMs with pretraining or fine-tuning. Figure 8 shows that well-trained LLMs after training/fine-tuning can exihibit the power-law Hessian structures, similarly to CNNs, while randomly initialized or pretrained models fail. Table 2 presents the KS test results on the power-law Hessian spectra across various LLMs, including GPT-2 series and TinyLlama-1b with LoRA adapter, in the case of pretraining or fine-tuning on various tasks.

The emergence of the power-law Hessian structure is evident throughout both the pretraining experiment (in Figure 9 and 10) and the fine-tuning experiment (in Figure 11). During pretraining, the power-law structure gradually develops as we optimize model parameters to capture the underlying structure of the training data. Similarly, during fine-tuning, the power-law structure adapts to reflect the specific characteristics of the target dataset. These findings, combined with empirical results from CNNs, suggest that the presence of the hessian Power-law structure is not confined to a specific

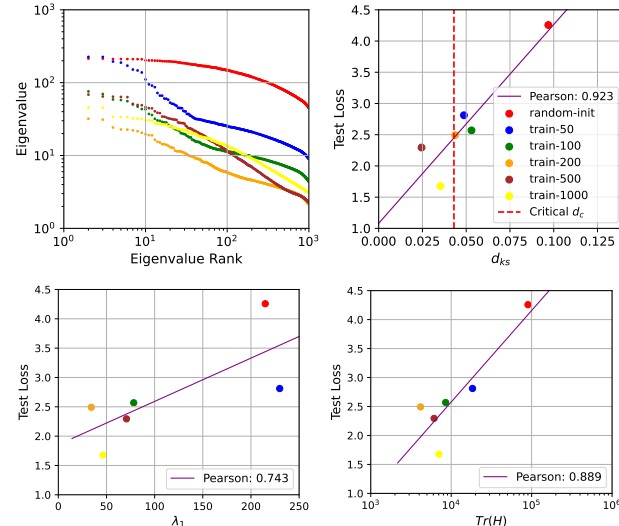

*Figure 10.* The KS distance $d_{ks}$ may serve as an effective predictor to language model's generalization abilities with a Pearson correlation coefficient up to 0.923. Model: GPT2-nano (trained from random initialization). Dataset: `Shakespeare`.

model architecture or limited to either vision (Figure 14) or text data (Figure 10). Further experiment results regarding different problem setups are provided in the Appendix E.

**2. Generalization Measure.** We studied how the generalization of LLMs closely relates to the power-law Hessian structure across different models and datasets.

We observed that for Figure 10 and Figure 11, minima sharpness represented by the largest eigenvalue $\lambda_1$ and Hessian trace $Tr(H)$ behaves poorly as a generalization measure for LLMs, contradicting conventional beliefs in deep learning (Jiang et al., 2019).

In the pretraining experiment of GPT2-nano, Figure 10 shows that pretraining can decrease the KS distance and test loss effectively. However, surprisingly, we discover that sharpness-based generalization measures become nearly useless for predicting generalization. In contrast, the KS distance can serve as an effective predictor to generalization ability with the Pearson correlation coefficient up to 0.923. Note that the KS distance $d_{ks}$ metric quantifies the adherence of a model's power-law spectral properties to those expected of a well-trained neural network.

In the fine-tuning experiment of GPT2-small in Figure 11, we notice similar observations. Again, the sharpness-based generalization measure fails, whereas the power-law goodness can work well. We can even see that the sharpness increases significantly during iterations 50-5000, whereas the test loss still drops quickly. Similar observation holds in Figure 9. In contrast, during all 5000 iterations, the test loss and the KS distance continuously drop synchronously

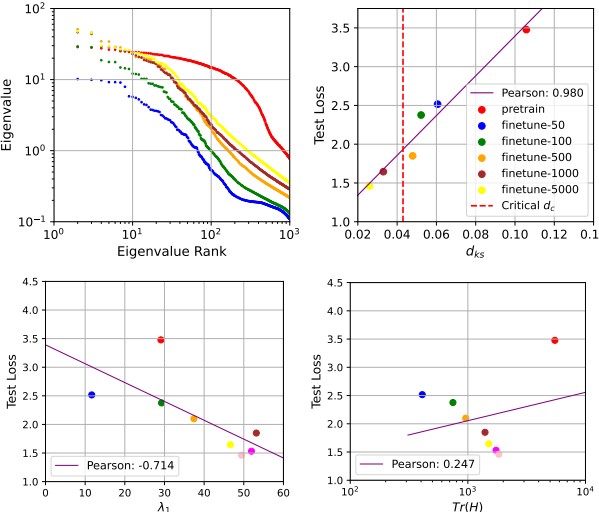

*Figure 11.* The power-law Hessian structure also emerged in fine-tuning tasks. The KS distance $d_{ks}$ predicts the model's generalization abilities with a strong Pearson correlation coefficient up to $0.980$, while the sharpness-based generalization measures obviously fail in predicting generalization of LLMs. Model: GPT2-small. Dataset: `Shakespeare`.

with the Pearson correlation coefficient up to 0.98, while the Pearson correlation coefficients for sharpness-based generalization measures are even only -0.714 and 0.247, which are harmful or nearly useless for predicting generalization.

We conjecture that the power-law Hessian structure and the minima's sharpness of the loss landscape can capture a model's generalization ability at different phases. Unlike CNNs, LLMs are often extremely over-parameterized and far from well-trained (e.g., many epochs). In the phase of staying far from minima, the power-law Hessian structure can better predict the generalization of LLMs. In contrast, as people usually train CNNs for many epochs, well-trained CNNs stay very close to minima. In the phase of staying close to minima, generalization can be captured by the minima's sharpness better, following conventional generalization theory. We believe a detailed generalization analysis for LLMs remains an open area for future research.

**3. Model Capacity and Scaling.** Scaling law predicts that the performance of LLMs typically follows power laws as we scale model parameters, training data, or computing (Bahri et al., 2024). We further investigate how the power-law Hessian structure depends on model scaling.

Figures 12 and 13 present the results of various pretraining checkpoint (step = 5000) and pretrained GPT-2 models trained and evaluated on OpenWebText. The results demonstrate that as model parameters increase, the power-law Hessian structure also becomes significantly more pronounced for both training and pretrained checkpoints, accompanied by improved model performance. The Pearson correlation

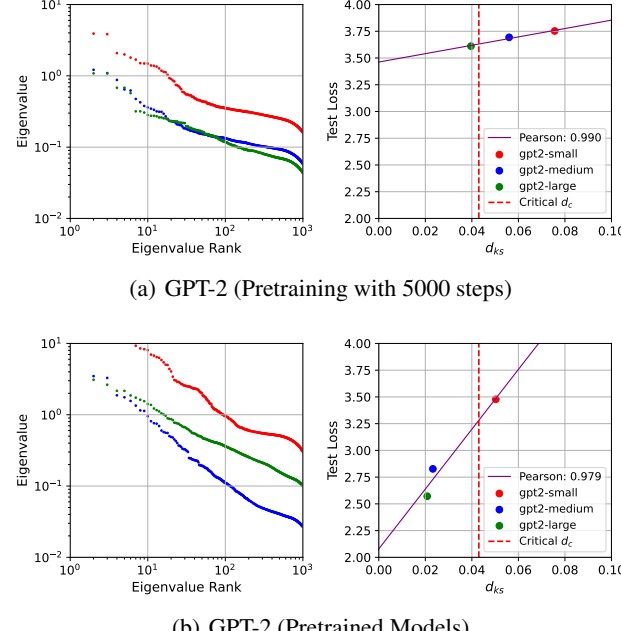

(a) GPT-2 (Pretraining with 5000 steps)

(b) GPT-2 (Pretrained Models)

*Figure 12.* The power-law spectrum holds across GPT-2 models with different model capacity. As we scale model parameters, the Hessian power-law structure goodness and test performance both improve. Subfigure (a) displays the eigenvalues of intermediate pretraining checkpoints at 5000 steps; Subfigure (b) displays the eigenvalues of the official pretrained GPT-2 models. Models: GPT2-{small, medium, large}. Dataset: `OpenWebText`.

coefficients are higher than 0.99 for the intermediate checkpoint and also high for the official pretrained models. This observation further supports that a scaling law perspective is reflected in the power-law Hessian structure of the LLM.

**4. Vision Transformer.** We also investigate the power-law Hessian structure of Vision Transformer on vision datsets. We analyze the pre-trained and fine-tuned ViT-base models (Dosovitskiy et al., 2021) on the CIFAR-100 dataset. Figure 14 shows that the power-law Hessian structure is absent in random ViT, and emerges in well-trained ViT models.

**5. Supplementary Results.** We present additional results and discussion, including layer-wise Hessian analysis and LoRA fine-tuning, in Appendix E. (1) In Figure 31, we observe that the power-law Hessian structure exists in various layers of LLMs. The power-law Hessian structure of the first and middle layers is often more pronounced than the last layer. (2) In Figure 33, we discover that while fine-tuning LLMs with LoRA can improve the performance of LLMs on a specific task, LoRA cannot significantly affect the Hessian structure of LLMs as full-parameter fine-tuning. The power-law Hessian structure does not emerge with LoRA fine-tuning. It suggests that full-parameter and LoRA fine-tuning may have significant generalization abilities.

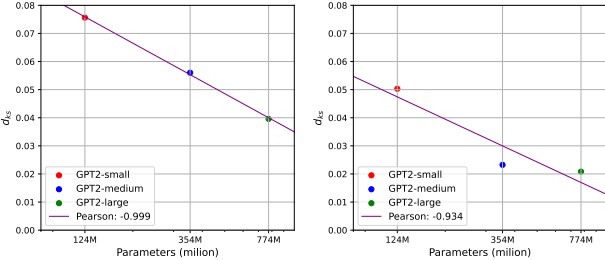

*Figure 13.* Scaling law of the power-law goodness for the Hessian structure. Larger models have a more precise power-law Hessian structure. Left: Intermediate GPT-2 pretraining checkpoints at 5000 steps; Right: Official pretrained GPT-2 models. Models: GPT2-{small, medium, large}. Dataset: `OpenWebText`.

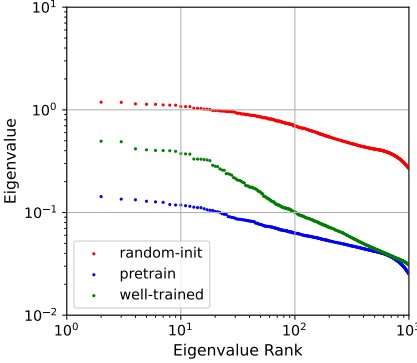

*Figure 14.* The power-law Hessian structure is absent in randomly initialized Vision Transformers and emerges with training. Model: ViT-base. Dataset: CIFAR-100.

## 5. Related Work

**The Hessian structure of DNNs.** A number of related works analyzed the spectral distribution of the Hessian in deep learning. Pennington & Bahri (2017) introduced an analytical framework from random matrix theory and reported that the shape of the spectrum depends strongly on the energy and the over-parameterization parameter, $\phi$, which measures the ratio of parameters to data points. However, Pennington & Bahri (2017) mainly evaluated single-hidden-layer networks, which limits the scope of the conclusion. A followup work (Pennington & Worah, 2018) focused on a single-hidden-layer neural network with Gaussian data and weights in the limit of infinite width. Obviously, its theoretical and empirical analysis is far from practical deep models. Jacot et al. (2019) analyzed the limiting spectrum of the Hessian in neural networks with infinite width. Fort & Scherlis (2019) analyzed the Hessian spectra of initialized neural networks. Papyan (2019) studied the three-level hierarchical structure and outliers in Hessian spectra. Singh et al. (2021) proved that the Hessian can be of very low rank for DNNs with linear activations. Liao & Mahoney

(2021) studied the Hessian spectra of more realistic nonlinear models. Kaur et al. (2023) studied the maximum Hessian eigenvalue and its relation to generalization. Dauphin et al. (2024) investigated the neglected Nonlinear Modeling Error (NME) matrix part of the Hessian and its influence on gradient penalties during training. While a number of works studied the Hessian spectra, they failed to empirically or theoretically discover the simple but important power-law structure, nor reveal its connection to generalization.

**Other power-law phenomena in deep learning.** Hestness et al. (2017) studied the power-law relation between model performance and model size as well as data size. Mahoney & Martin (2019) reported that the elements of weight matrices may exhibit power-law heavy tails and studied the trends of spectral decay. Lee et al. (2020); Velikanov & Yarotsky (2021) reported the power-law decaying eigenvalues in kernel methods. Agrawal et al. (2022) studied the eigenspectrum decaying of feature covariance with a theoretical analysis of linear regression. Xie et al. (2023a) studied the heavy-tailed structure of stochastic gradient covariance. However, none of them reported the overlooked power-law Hessian structure of DNNs.

## 6. Conclusion

While the Hessian of the deep loss landscape matters to optimization and generalization of deep learning, the statistical structure of the Hessian is still largely overlooked by previous studies. To the best knowledge, we are the first to report the overlooked power-law Hessian structure in deep learning as well as formal statistical tests on the power-law Hessian spectra. We provide a novel maximum-entropy interpretation and explain why the learning space may be low-dimensional and robust. While the main limitation of our work is that we cannot include those state-of-the-art LLMs in our experiments due to the extremely large memory and computational cost of Hessian analysis, our work still goes much further beyond those previous qualitative studies. The power-law Hessian structure provides a useful and novel perspective to reveal and analyze multiple novel behaviors of deep learning on optimization, generalization, and over-parameterization. We particularly discovered that while conventional sharpness-based generalization measures are considered nice generalization predictors of CNNs, they often completely fail to predict the generalization of LLMs. Instead, the power-law goodness of the Hessian structure often correlates better with generalization on many deep learning occasions, while we also observe that it is not a robust generalization measure. This suggests that generalization theory and measures of LLMs lacks more exploration and rethinking. We believe that our work will further inspire theories and empirical advances toward a deeper understanding of DNNs, loss landscape, and Hessian.

## Acknowledgement

This work was sponsored by Doubao Large Model Fund of ByteDance, Natural Science Foundation of China (No. 12305052), Research Grants Council of Hong Kong (Nos. 22302723 and SRFS2324-2S05), Hong Kong Baptist University's funding support (RC-FNRA-IG/22-23/SCI/03).

## Impact Statement

This paper presents work whose goal aims at understanding the foundation of deep learning. While it may have many potential societal consequences, we think none of them must be specifically discussed here.

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

## A. Proof of Theorem 1

*Proof.* Considering the principle of maximum entropy with the two kinds of entropy, we need to maximize the total entropy with the spectral density normalization constraint

$$S_{\text{total}} = -\int p(\lambda) \log p(\lambda) d\lambda - \beta_{\text{vol}} \int p(\lambda) \log \lambda d\lambda \tag{8}$$
$$- \beta_{\text{norm}} \left( \int p(\lambda) d\lambda - 1 \right),$$

where $S_{\text{total}} = S_{\text{p}} + \beta_{\text{vol}} S_{\text{vol}}$ and $\beta_{\text{norm}}$ is a Lagrange multiplier. To find the optimal distribution $p^\star(\lambda)$ that maximizes the total entropy, we require the following

$$\frac{\partial S_{\text{total}}}{\partial p(\lambda)} = -\log p(\lambda) - \beta_{\text{vol}} \log \lambda - \beta_{\text{norm}} = 0. \tag{9}$$

Thus, the optimal distribution $p^\star(\lambda)$ can be solved as

$$p^\star(\lambda) = e^{-\beta_{\text{norm}}} \lambda^{-\beta_{\text{vol}}}. \tag{10}$$

$\square$

## B. Robust and Low-Dimensional Learning Space

At first, we denote the ordered eigengap as $\delta_k = \lambda_k - \lambda_{k+1}$, which means the difference between two neighbored eigenvalues. According to (2), we have

$$\delta_k = \text{Tr}(H) Z_d^{-1} (k^{-\frac{1}{\beta-1}} - (k+1)^{-\frac{1}{\beta-1}}) \tag{11}$$
$$= \lambda_k \left[ 1 - (\frac{k}{k+1})^s \right]. \tag{12}$$

It shows that the eigengaps of Hessians also approximately exhibit a power-law distribution

$$\delta_k = \text{Tr}(H) Z_d^{-1} (k+1)^{-(s+1)} \tag{13}$$

under the approximation $s \approx 1$. The power exponent $s + 1$ is larger than the one in (2) by 1.

The phenomenon of low-dimensional learning subspace was empirically reported by Gur-Ari et al. (2018). Ghorbani et al. (2019) also investigated and reported that large isolated eigenvalues quickly appear in the spectrum during the optimization process, along with a surprising concentration of the gradient in the corresponding eigenspace. However, they did not theoretically explain this phenomenon. Xie et al. (2021b) theoretically demonstrated that learning dynamics mainly happens along those principal eigenvectors of Hessian corresponding to large eigenvalues. Does the phenomenon theoretically depend on the Hessian structure? Our answer is yes. In the following part, we will demonstrate why the eigengaps of the Hessian $H$ may naturally lead to the phenomenon that learning dynamics mainly takes place in a low-dimensional space during the entire training process.

Previous papers only reported that top eigenvalues of Hessian are significantly larger than other tailed ones but did not touch how top Hessian eigengaps dominate other tailed ones in deep learning. However, top large eigenvalues do not imply their eigengaps are relatively large, too. Fortunately, we theoretically and empirically demonstrate that, as the rank index increases, both eigenvalues and eigengaps decay following power laws. Moreover, the eigengaps decay faster than eigenvalues. The theoretical implication behind the power-law eigengaps actually matters to deep learning dynamics.

We directly apply Theorem 2, a useful variant of Davis-Kahan Theorem (Yu et al., 2015), to the Hessian in deep learning.

**Theorem 2** (A useful variant of Davis-Kahan Theorem (Yu et al., 2015)). *Suppose the true Hessian is $H$, the perturbed Hessian is $\tilde{H} = H + \epsilon M$, the $i$-th eigenvector of $H$ is $u_i$, and its corresponding perturbed eigenvector is $\tilde{u}_i$. Under the conditions of the Davis-Kahan Theorem, we have*

$$\sin\langle u_k, \tilde{u}_k \rangle \leq \frac{2\epsilon \|M\|_{op}}{\min(\lambda_{k-1} - \lambda_k, \lambda_k - \lambda_{k+1})},$$

*where $\|M\|_{op}$ is the operator norm of $M$.*

Given the power-law eigengaps in Equation (13), the upper bound of eigenvector robustness can be written as

$$\sup \sin\langle u_k, \tilde{u}_k \rangle = \frac{2\epsilon \|M\|_{op}(k+1)^{s+1}}{\lambda_1},\tag{14}$$

which is relatively tight for top dimensions and very loose for tailed dimensions. A similar conclusion also holds given Equation (11). This indicates that non-top eigenspace can be highly unstable during training, because $\delta_k$ can decay to nearly zero for a large $k$. To the best of our knowledge, we are the first to demonstrate that the robustness of low-dimensional learning space directly depends on the eigengaps of the Hessian $H$.

# C. Experimental Settings

**Computational environment.** The image classification experiments are conducted on a computing cluster with NVIDIA® V100/H800 GPUs and Intel® Xeon® CPUs.

## C.1. Image Classification:

### C.1.1. MODELS, DATASETS, AND OPTIMIZERS

**Models:** LeNet (LeCun et al., 1998), Fully Connected Networks (FCN), ResNet18 (He et al., 2016) and Vision Transformer (Dosovitskiy et al., 2021). Particularly, we used one-layer FCN, two-layer FCN, four-layer FCN, which have 100 neurons for each hidden layer and use ReLu activations.

**Datasets:** MNIST (LeCun, 1998), Fashion-MNIST (Xiao et al., 2017), CIFAR-10/100 (Krizhevsky & Hinton, 2009), and non-image Avila (De Stefano et al., 2018).

**Optimizers:** SGD, Vanilla SGD, Adam (Kingma & Ba, 2015), AMSGrad (Reddi et al., 2019), AdaBound (Luo et al., 2019), Yogi (Zaheer et al., 2018), RAdam (Liu et al., 2019), Adai (Xie et al., 2022), PNM (Xie et al., 2021c), Lookahead (Zhang et al., 2019), and DiffGrad (Dubey et al., 2019).

### C.1.2. IMAGE CLASSIFICATION ON MNIST AND FASHION-MNIST

**Data Preprocessing For MNIST and Fashion-MNIST:** We perform the common per-pixel zero-mean unit-variance normalization.

**Hyperparameter Settings:** We select the optimal learning rate for each experiment from $\{0.0001, 0.001, 0.01, 0.1, 1, 10\}$ for SGD and use the default learning rate for adaptive gradient methods. In the experiments on MNIST and Fashion-MNIST: $\eta = 0.1$ for SGD, Vanilla SGD, Adai, PNM, and Lookahead; $\eta = 0.1$ for Vanilla SGD; $\eta = 0.001$ for Adam, AMSGrad, AdaBound, Yogi, RAdam, and DiffGrad.

We train neural networks for 50 epochs on MNIST and 200 epochs on Fashion-MNIST. For the learning rate schedule, the learning rate is divided by 10 at the epoch of $40\%$ and $80\%$. The batch size is set to 128 for MNIST and Fashion-MNIST, unless we specify it otherwise.

The strength of weight decay defaults to $\lambda = 0.0005$ as the baseline for all optimizers unless we specify it otherwise.

We set the momentum hyperparameter $\beta_1 = 0.9$ for SGD and adaptive gradient methods which involve in Momentum. As for other optimizer hyperparameters, we apply the default settings directly.

### C.1.3. IMAGE CLASSIFICATION ON CIFAR-10 AND CIFAR-100

**Data Preprocessing For CIFAR-10 and CIFAR-100:** We perform the common per-pixel zero-mean unit-variance normalization, horizontal random flip, and $32 \times 32$ random crops after padding with 4 pixels on each side.

**Hyperparameter Settings:** We select the optimal learning rate for each experiment from $\{0.0001, 0.001, 0.01, 0.1, 1, 10\}$ for SGD and use the default learning rate for adaptive gradient methods. In the experiments on CIFAF-10 and CIFAR-100: $\eta = 1$ for Vanilla SGD, Adai, and PNM; $\eta = 0.1$ for SGD (with Momentum) and Lookahead; $\eta = 0.001$ for Adam, AMSGrad, AdaBound, Yogi, RAdam, and DiffGrad. For the learning rate schedule, the learning rate is divided by 10 at the epoch of $\{80, 160\}$ for CIFAR-10 and $\{100, 150\}$ for CIFAR-100, respectively. The batch size is set to 128 for both CIFAR-10 and CIFAR-100, unless we specify it otherwise.

The strength of weight decay is default to $\lambda = 0.0005$ as the baseline for all optimizers unless we specify it otherwise. Xie et al. (2023b) found that popular optimizers with $\lambda = 0.0005$ often yields test results than $\lambda = 0.0001$ for training CNNs on CIFAR-10 and CIFAR-100.

We set the momentum hyperparameter $\beta_1 = 0.9$ for SGD with Momentum. As for other optimizer hyperparameters, we apply the default hyperparameter settings directly.

### C.1.4. LEARNING CNNs WITH NOISY LABELS

We trained LeNet via SGD (with Momentum) on corrupted MNIST with various (asymmetric) label noise. We followed the setting of Han et al. (2018) for generating noisy labels for MNIST. The symmetric label noise is generated by flipping every label to other labels with uniform flip rates $\{40\%, 80\%\}$. In this paper, when we talk about label noise, we mean symmetric label noise.

We also randomly shuffle the labels of MNIST to produce MNIST with random labels, which has little knowledge behind the pairs of instances and labels.

### C.1.5. IMAGE CLASSIFICATION ON VISION TRANSFORMER

**Data Preprocessing For CIFAR-100:** We perform resizing of CIFAR-100 images to $224 \times 224$ pixels for compatibility during training. We then perform horizontal random flip and the common per-pixel zero-mean unit-variance normalization.

**Hyperparameter Settings:** We fine-tuned the pretrained Vision Transformer (ViT-Base) on the CIFAR-100 dataset, following the experimental setup of Dosovitskiy et al. (2021). We trained the pretrained ViT with a total batch size of 4096 for 200 steps using a learning rate of 1e-4 with linear decay. We employed the Adam optimizer with betas set to (0.9, 0.999) and applied a weight decay strength of 0.1.

**Hessian Spectra Computation:** We utilized the Stochastic Lanczos Quadrature (SLQ) algorithm implementation from Yao et al. (2020) for computing the Hessian eigenvalues for ViT, incorporating modifications for improved computational and memory efficiency. To further reduce the computational cost of SLQ, we sampled 1000 samples in each run and approximate the model's Hessian on the target dataset. For all Hessian experiments on ViT, 3,000 eigenvalues were computed across three repeated SLQ runs to mitigate bias, with only the top 1,000 eigenvalues displayed and used for evaluation, consistent with the experiment setup for LLMs.

### C.2. Text Generation:

### C.2.1. MODELS, DATASETS AND OPTIMIZERS

**Models:** GPT-2 (Radford et al., 2019), Tinyllama (Zhang et al., 2024a). We use the code base of NanoGPT (Karpathy, 2022) for reproducing all GPT-2 models with pretraining and fine-tuning experiments.

**Datasets:** OpenWebText (Gokaslan et al., 2019), Shakespeare (Karpathy, 2015) (processed at the character level), and MathQA (Amini et al., 2019).

**Optimizer:** AdamW (Loshchilov, 2017) is used for training and fine-tuning of all language models utilized in this study.

### C.2.2. IMPLEMENTATION DETAILS ON HESSIAN SPECTRA COMPUTATION

To enable efficient computation of the Hessian spectra for LLMs, we applied the Stochastic Lanczos Quadrature algorithm implementation provided by Yao et al. (2020) with alternations for computational and memory speedups. As computation cost of SLQ is unaffordable on large-scale text dataset as discussed in Zhang et al. (2024b), we applied the same batch sampling trick and used a fixed batch size of 256 in all of our text generation experiments to approximate model's Hessian evaluated on the target dataset. For all experiments conducted on LLMs, 3,000 eigenvalues were computed across 3 repeated SLQ runs to minimize bias; Only the top 1,000 eigenvalues were displayed and used for evaluation in all Figures.

### C.2.3. TRAINING CONFIGURATIONS

- **GPT2-nano trained on Shakespeare.** We utilized a smaller 'baby GPT' model with 6 layers, 6 attention heads, an embedding size of 384, and 11M parameters provided in NanoGPT. We used the AdamW optimizer with a fixed learning rate = $6 \times 10^{-4}$. We used a batch size = 65,536 tokens and weight decay = 0.1 for a total of 1,000 steps.

*Table 3.* The Kolmogorov-Smirnov statistics of various optimizers for training LeNet on CIFAR-10.

| Training | $d_{\text{ks}}$ | $d_{\text{c}}$ | Power-Law | $\hat{\beta} \pm \sigma$ | $\hat{s}$ |
|---|---|---|---|---|---|
| Random | 0.0663 | 0.0430 | No | | |
| SGD | 0.0279 | 0.0430 | Yes | $1.968 \pm 0.031$ | 1.033 |
| Vanilla SGD | 0.0276 | 0.0430 | Yes | $1.935 \pm 0.030$ | 1.069 |
| Adam | 0.0269 | 0.0430 | Yes | $1.806 \pm 0.025$ | 1.241 |
| AMSGrad | 0.0232 | 0.0430 | Yes | $1.786 \pm 0.025$ | 1.271 |
| AdaBound | 0.0297 | 0.0430 | Yes | $1.901 \pm 0.028$ | 1.110 |
| Yogi | 0.0184 | 0.0430 | Yes | $1.806 \pm 0.025$ | 1.241 |
| RAdam | 0.0163 | 0.0430 | Yes | $1.733 \pm 0.023$ | 1.363 |
| Adai | 0.0310 | 0.0430 | Yes | $1.918 \pm 0.029$ | 1.090 |
| PNM | 0.0347 | 0.0430 | Yes | $1.911 \pm 0.029$ | 1.098 |
| Lookahead | 0.0358 | 0.0430 | Yes | $1.964 \pm 0.030$ | 1.037 |
| DiffGrad | 0.0303 | 0.0430 | Yes | $1.803 \pm 0.024$ | 1.236 |

*Table 4.* The Kolmogorov-Smirnov statistics of the Hessian spectra for various batch sizes.

| Dataset | Model | Training | Batch Size | $d_{\text{ks}}$ | $d_{\text{c}}$ | Power-Law | $\hat{\beta} \pm \sigma$ | $\hat{s}$ |
|---|---|---|---|---|---|---|---|---|
| MNIST | LeNet | SGD | $B = 128$ | 0.00900 | 0.0430 | Yes | $1.991 \pm 0.031$ | 1.009 |
| MNIST | LeNet | SGD | $B = 512$ | 0.00787 | 0.0430 | Yes | $1.894 \pm 0.028$ | 1.119 |
| MNIST | LeNet | SGD | $B = 640$ | 0.0125 | 0.0430 | Yes | $1.838 \pm 0.027$ | 1.194 |
| MNIST | LeNet | SGD | $B = 768$ | 0.278 | 0.0430 | No | | |
| MNIST | LeNet | SGD | $B = 1024$ | 0.129 | 0.0430 | No | | |
| MNIST | LeNet | SGD | $B = 16384$ | 0.249 | 0.0430 | No | | |
| MNIST | LeNet | SGD | $B = 32768$ | 0.201 | 0.0430 | No | | |
| MNIST | LeNet | SGD | $B = 50000$ | 0.139 | 0.0430 | No | | |
| MNIST | LeNet | SGD | $B = 60000$ | 0.0936 | 0.0430 | No | | |

- **GPT2-{small, medium, large} pretrained on OpenWebText.** We used the AdamW optimizer with a learning rate = $6 \times 10^{-4}$, incorporating 2,000 warmup steps and a decay schedule down to $6 \times 10^{-5}$, following the experiment setup in NanoGPT. We used a batch size = 65,536 tokens and weight decay = 0.1 for a total of 500,000 steps.

- **GPT2-{small, medium, large} fine-tuned on OpenwebText / Shakespeare.** We used the AdamW optimizer with a learning rate = $1 \times 10^{-5}$, incorporating 1,000 warmup steps and a decay schedule down to $1 \times 10^{-6}$. We used a batch size = 32,768 tokens and weight decay = 0.1 for a total of 5,000 steps.

- **TinyLlama fine-tuned on MathQA.** We fine-tuned the TinyLlama model with LoRA adapters, with rank = 16, alpha = 32, dropout = 0.1 as specified in the original LoRA experiment (Hu et al., 2021). We used the AdamW optimizer with a learning rate = $1 \times 10^{-4}$, incorporating 500 warmup steps and a decay schedule down to $1 \times 10^{-5}$. We used a batch size = 32 and weight decay = 0.1 for 1,000 steps.

## D. Supplementary Experiment Results of Convolutional Neural Networks

**Three phases in large-batch training.** In this section, we will further discuss the three phase changes in large-batch image classification training. We followed the same experiment setup in Table D as stated in Appendix C.1.2. First, in Phase I ($B \leq 640$), moderately large-batch ($B = 512$) training indeed finds sharper minima than small-batch ($B = 128$) training, while the power-law spectrum still holds well. Power laws may guarantee that the top eigenvalues of large-batch trained networks are all larger than the corresponding eigenvalues of small-batch trained networks. The main challenge of large-batch training in Phase I is consistent with the common belief that large-batch training suffers from sharp minima and, thus, leads to bad generalization (Hoffer et al., 2017). The minima sharpness measured by $\hat{s}$ increases with the batch size.

Second, in Phase II ($768 \leq B \leq 50000$), the spectrum of large-batch ($B = 1024$) trained networks does not exhibit power laws but is visually similar to the spectrum of underparameterized models in Figure 5. In Phase II, large-batch trained overparameterized models behave like underparameterized models from a spectral perspective, and, thus, can lead to bad

generalization. The phase transition from Phase I to Phase II occurs in a narrow range of $640 < B < 768$, which is visually observable in Figure 7a and statistically observable in Table D.

Third, in Phase III ($B \sim 60000$), extremely large-batch training ($B = 60000$) cannot optimize the training loss well or find the Hessian spectra similarly to random initialized neural networks. Phase III indicates that, sometimes, bad convergence rather than sharp minima can become the main performance bottleneck in large-batch training when the batch size is too large.

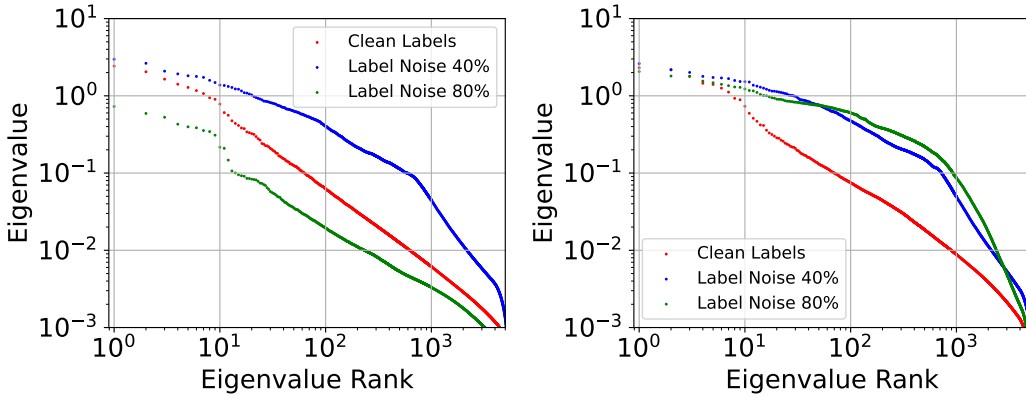

*Figure 15.* The spectrum in the presence of noisy labels. Top: MNIST Trainset. Bottom: MNIST Testset.

Our Hessian spectra analysis discovery differs from traditional beliefs that different phases exist in training. In phase 2, the Hessian eigenvalue increases and breaks the power-law structure, while the model's performance is much superior compared to untrained neural networks. We may leave a deeper investigation for future work.

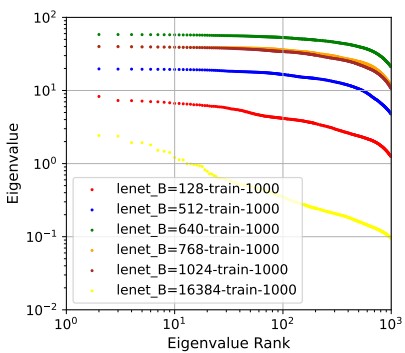

*Table 5.* The Kolmogorov-Smirnov statistics of the Hessian spectra for various batch sizes under the same number of training iterations.

| Dataset | Model | Training | Batch Size | $d_{\mathrm{ks}}$ | $d_{\mathrm{c}}$ | Power-Law |
|---------|-------|----------|------------|---------|--------|-----------|
| MNIST | LeNet | SGD | $B = 128$ | 0.0527 | 0.0430 | No |
| MNIST | LeNet | SGD | $B = 512$ | 0.0934 | 0.0430 | No |
| MNIST | LeNet | SGD | $B = 640$ | 0.1138 | 0.0430 | No |
| MNIST | LeNet | SGD | $B = 768$ | 0.1242 | 0.0430 | No |
| MNIST | LeNet | SGD | $B = 1024$ | 0.0999 | 0.0430 | No |
| MNIST | LeNet | SGD | $B = 16384$ | 0.0271 | 0.0430 | Yes |

*Figure 16.* The Hessian spectrum for various batch sizes under the same number of training iterations.

**Clean and Random Labels.** We presented the spectrum of learning with clean labels and random labels in Figure 17. The number of top outliers obviously increases, because random labels make the dataset more complex. However, even if the pairs of instances and labels have little knowledge, we still observe the power-law spectrum after the dozens of top outliers. This may suggest that, even if the labels are random, neural networks can still learn useful knowledge from the instances only.

**Overfitting and Noisy Labels.** As DNNs overfit noisy labels easily, previous papers choose learning with noisy labels as an important setting for evaluating overfitting and generalization (Han et al., 2020; Xie et al., 2021a; He et al., 2022). Figure 15 shows that overfitting label noise makes the Hessian spectra less power-law on both the corrupted training dataset and the clean test dataset. In contrast, in the absence of noisy labels, the power-law spectra exist on both the training dataset and the test dataset.

**Transferability.** In transfer learning, people believe that a model pretrained on one dataset may learn useful representations

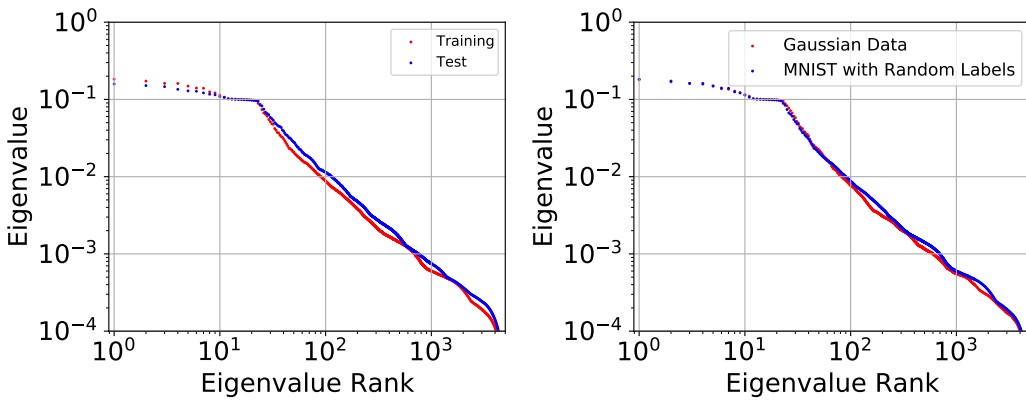

*Figure 17.* (a) The spectrum of LeNet on (training and test) MNIST with randomly shuffled labels. (b) The spectrum of LeNet on Gaussian data with randomly shuffled labels is highly similar to that MNIST with randomly shuffled labels.

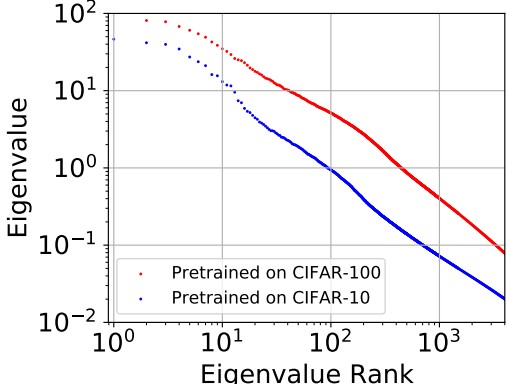

*Figure 18.* Transferability of the power-law Hessian structure. A LeNet which is pretrained on CIFAR-100 may still exhibit power-law Hessian spectra when evaluated on CIFAR-10.

for relevant datasets or downstream tasks. When we pretrain a model on CIFAR-100 and evaluate its Hessian spectrum on CIFAR-10, we surprisingly discover that the power-law Hessian structure successfully transfers. This may measures the usefulness of learned representations.

**Power Iteration vs. Lanczos Algorithm.** We compared the spectra computed via Power Iteration Algorithm and Lanczos Algorithm in Figure 19. It shows the top eigenvalues estimated via Power Iteration Algorithm are highly consistent with the top eigenvalues via the Lanczos Algorithm. It also demonstrates that the power-law spectrum is caused by the properties of deep learning rather than the stochasticity of Lanczos Algorithm.

We presented the power-law eigengaps on Fashion-MNIST in Figure 21. It shows that the power-law eigengaps on Fashion-MNIST are highly consistent with the power-law eigengaps on MNIST.

We presented the power-law spectrum of the covariance matrix of stochastic gradient noise of FCN on MNIST in Figure 27. As the inverses of the power-law variables are power-law, the covariance spectrum shows heavy-tail properties. It demonstrates that the heavy-tail property belongs to deep neural networks rather than SGD itself.

**Avila Dataset** We presented the power-law spectrum of two-layer FCN on Avila Dataset in Figure 20. It shows that the power-law spectrum of neural networks may also generally exist in non-convolution neural networks trained on a non-image dataset. Particularly, we note that the Avila Dataset has only ten attributes, including intercolumnar distance, upper margin, lower margin, exploitation, row number, modular ratio, interlinear spacing, weight, peak number, and modular ratio/ interlinear spacing. These attributes are essentially different from the pixels in image datasets.

**Supplementary Empirical Results on CIFAR-10.** The spectra of LeNet on CIFAR-10 trained via various optimizers

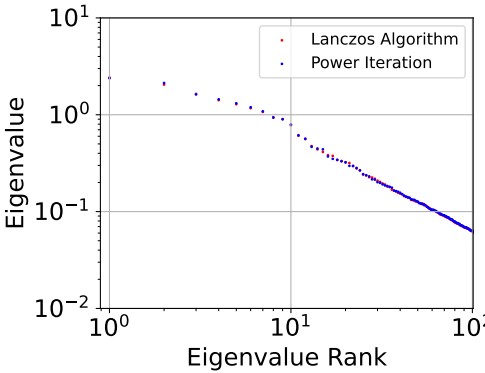

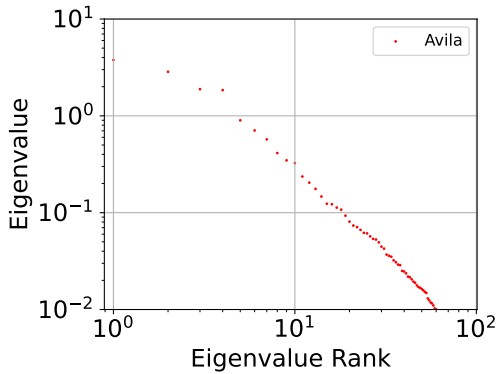

*Figure 19.* The spectrum via Power Iteration Algorithm is highly consistent with the spectrum via Lanczos Algorithm. It also shows that the power-law spectrum is caused by the properties of deep learning rather than the stochasticity of Lanczos Algorithm.

*Figure 20.* The spectrum of FCN on Avila Dataset. It shows that the power-law spectrum of neural networks may also exist in non-image datasets.

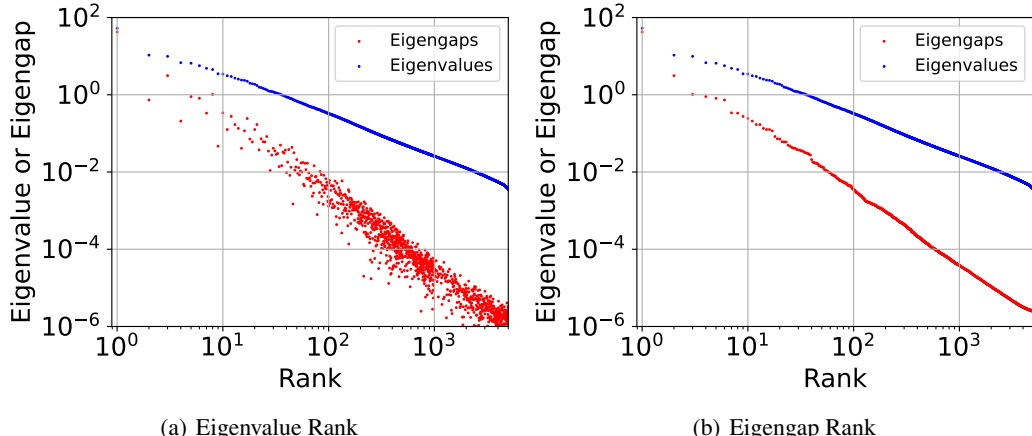

(a) Eigenvalue Rank

(b) Eigengap Rank

*Figure 21.* The power-law Hessian eigengaps in deep learning. Model: LeNet. Datset: Fashion-MNIST. Subfigure (a) displayed the eigengaps by original rank indices sorted by eigenvalues. Subfigure (b) displayed the eigengaps by rank indices re-sorted by eigengaps.

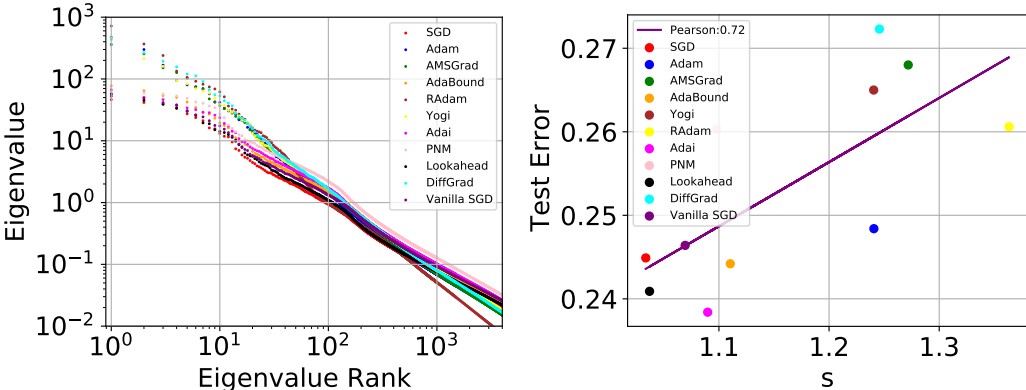

*Figure 22.* The power-law spectra hold across optimizers. Moreover, the slope magnitude $\hat{s}$ is an indicator of minima sharpness and a predictor of test performance. Model:LeNet. Dataset: CIFAR-10.

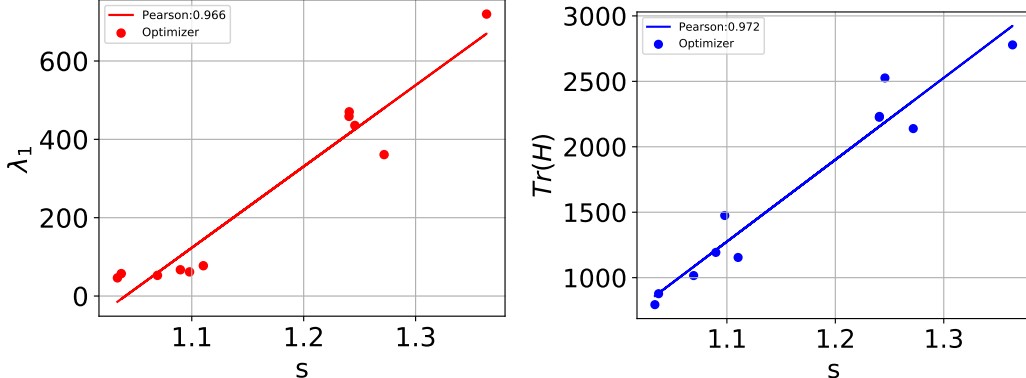

*Figure 23.* The slope magnitude $\hat{s}$ closely correlates with the largest Hessian eigenvalue and the Hessian trace. Model:LeNet. Dataset: CIFAR-10.

are showed in Figure 22. Figures 4 and 23 shows that the slope magnitude $\hat{s}$ closely correlates with the largest Hessian eigenvalue and the Hessian trace.

We presented the power-law spectra of ResNet18 on CIFAR-10 in Figure 24. It shows that the power-law spectra hold for ResNet, a representative of the modern neural network architectures, as well as simple CNNs/FCNs. Due to the GPU memory limit, we may only display the top 50 eigenvalues for ResNet18. However, the KS test still supports accepting the power-law hypothesis.

We report the spectra of large-batch trained ResNet18 on CIFAR-10 in Figure 25. It indicates that the phase transition behaviors of the spectra with respect to batch size generally exist. However, it seems that Phase II and Phase III merge into one phase for ResNet18 on CIFAR-10.

Figure 26 shows that the small width of neural networks may also break the power-law spectrum like small depth. This also supports that overparameterization or large model capacity is necessary for the power-law spectrum.

**Rethinking the heavy-tail phenomenon in SGD.** The heavy-tail property of SGD has been a hot and arguable topic recently (Simsekli et al., 2019; Panigrahi et al., 2019; Gurbuzbalaban et al., 2021; Hodgkinson & Mahoney, 2021; Xie et al., 2021b; Li et al., 2021). Note that the power-law distribution is one of the most common heavy-tail distributions in the real world. We argue that the arguable heavy-tail property of SGD may depend on the power-law Hessian spectrum rather than SGD itself, as gradient noise covariance critically depends on the Hessian. We present the power-law spectra of gradient noise covariance in Figure 27.

We specifically study the Hessian spectrum of Linear Neural Networks (LNNs) which has no ReLu and BatchNorm. Figure

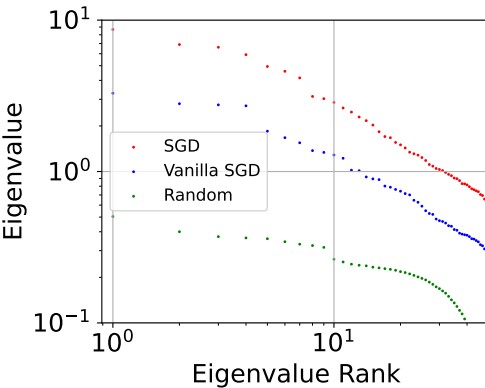

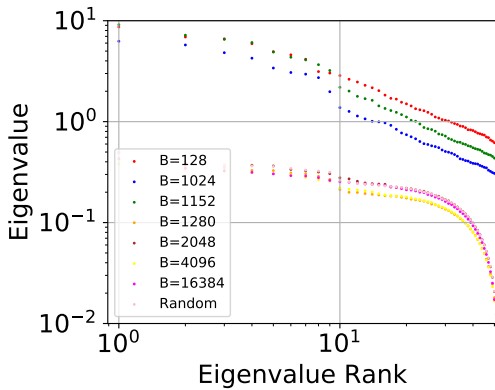

*Figure 24.* The power-law spectra of ResNet18 on CIFAR-10. It shows that the power-law spectrum of neural networks may also exist in modern neural network architectures (ResNet) as well as simple CNNs/FCNs.

*Figure 25.* Batch size matters to the spectrum. Model:ResNet-18. Dataset: CIFAR-10. The sharp phase transition occurs in $1152 < B < 1280$.

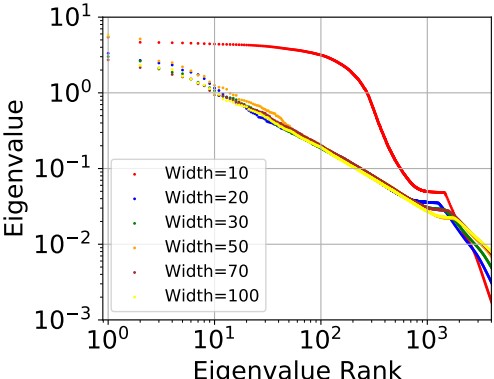

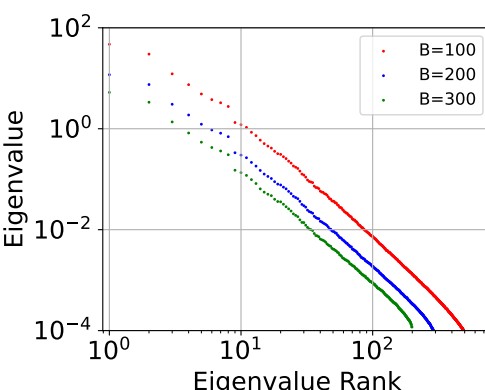

*Figure 26.* The spectra are not power-law for neural networks with a small width($\sim 10$), but gradually become more power-law (more straight in the log-log plot) as the width increases. This may also suggest that the power-law spectrum depends on model capacity. Model: Two-layer FCN. Dataset: MNIST.

*Figure 27.* The power-law spectrum of gradient noise covariance exists in deep learning for various batch sizes. Model: Fully Connected Network(FCN). Dataset: MNIST.

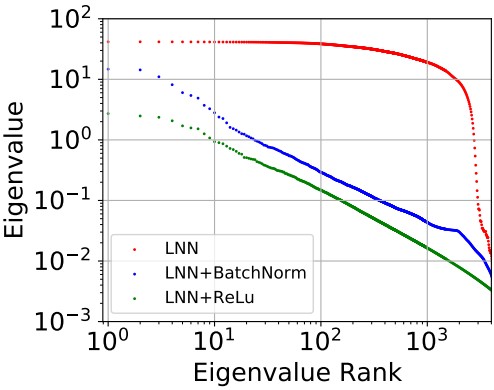
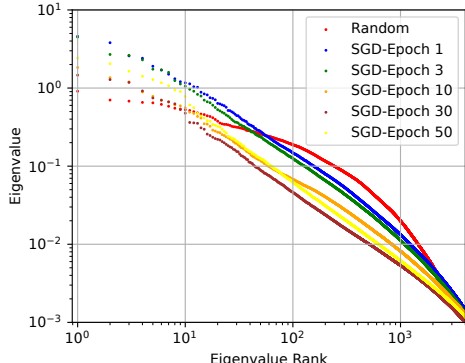

*Figure 28.* The spectrum of LNN with or without BatchNorm and ReLu.

*Figure 29.* The spectrum of LeNets trained with various epochs.

29 shows that the spectrum of LNNs is not power-law but more like spectra of underparameterized models. It may indicate that nonlinearity and BatchNorm both help improve model capacity. LNNs equipped with ReLu or BatchNorm immediately recover the power-law spectra again.

## E. Supplementary Experiment Results of Large Language Models

### E.1. Power-law Hessian Eigengaps in LLM

In this subsection, we report the power-law spectrum in Hessian eigengaps of fine-tuned GPT2-small. As previous results of LeNet (Figure 2, 21), GPT2-small as a language model, demonstrated that learning dynamics similarly takes place in a low-dimensional learning subspace as proved in Appendix B.

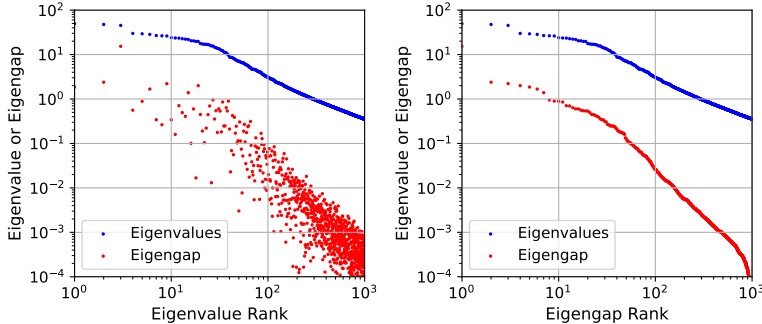

*Figure 30.* The power-law Hessian eigengaps. Model: GPT2-small. Dataset: `Shakespeare`. Subfigure (a) displayed the eigengaps by original rank indices sorted by eigenvalues. Subfigure (b) displayed the eigengaps by rank indices re-sorted by eigengaps.

### E.2. Layer-wise Power-law Spectra

Due to computational constraints, we reported the Hessian spectra only using the parameters of the last layer in previous sections. In this section, we will present and discuss the power-law spectrum at different layers of the fine-tuned GPT2-small model. Figure 31 presented the power-law spectrum at $2, 5, 8, 11^{th}$ layers of GPT2-small checkpoints fine-tuned on the `Shakespeare` dataset. The phenomenon is clear: the power-law spectrum forms early in training, particularly in the earlier layers, and the eigenvalue magnitudes decrease across layers.

### E.3. Fine-tuning with Low-dimensional Adaptations (LoRA)

In this section, we report the Hessian eiganvalues of the low dimensional adapters (LoRA (Hu et al., 2021)) for efficient fine-tuning of the pretrained TinyLlama-1B model on the MathQA dataset. The top 1000 eigenvalues are computed for

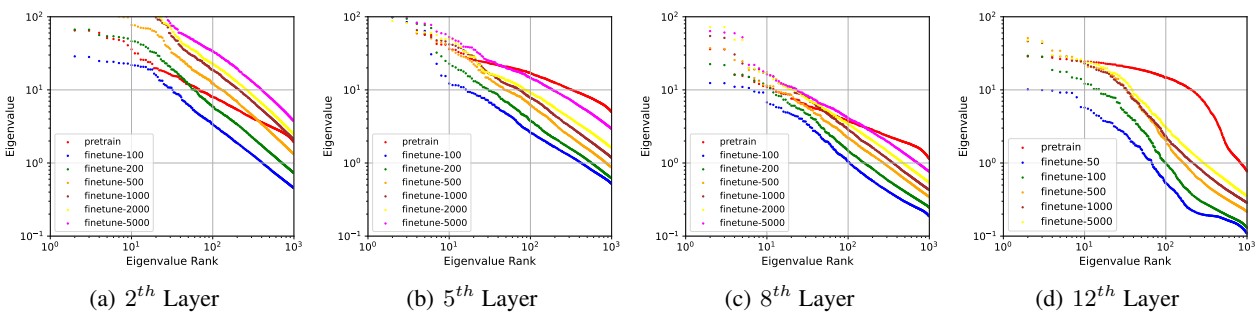

*Figure 31.* Power-law spectrum at different layers and checkpoints of the GPT2-small model. Dataset: `Shakespeare`.

the Hessian of all LoRA layers are displayed, and we may observe that a hessian Power-law structure is formed after 500 fine-tuning steps.

We then discuss how generalization of the fine-tuned LoRA adapters relate to the power-law Hessian structure. In Figure 32, the KS distance ($d_{ks}$) calculated from the Hessian spectra of the adapters, initially increases during the first 100 steps of fine-tuning before decreasing alongside the test loss. Although the largest eigenvalue $\lambda_1$ increases at step 50 compared to its random initialization, the Hessian trace $Tr(H)$ consistently decreases throughout fine-tuning. This behavior suggests that for pretrained models with good proficiency, the power-law spectra may diverge from expected patterns, and the Hessian trace $Tr(H)$ may be a more reliable indicator of generalization capability according to Figure 32.

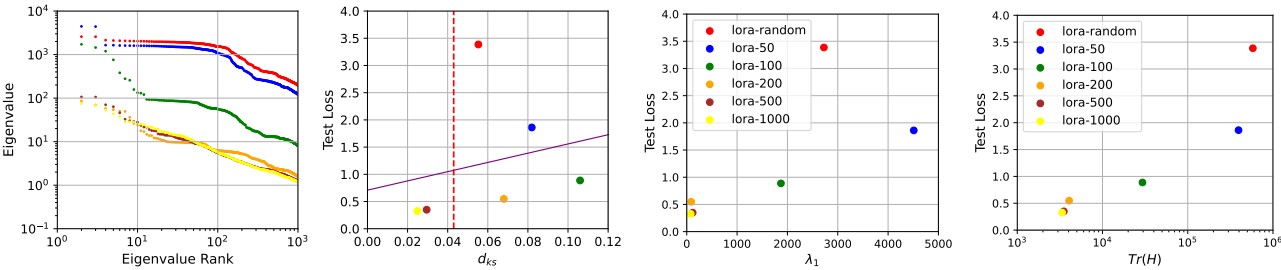

*Figure 32.* The behaviour of the Hessian spectrum of all LoRA adapters aligns similarity to the original model. LoRA adapters are randomly initialized before fine-tuning. Model: LoRA Layers (adapted on TinyLlama). Dataset: `MathQA`.

We further computed the Hessian spectrum of the pretrained last layer of TinyLlama-base merged with the fine-tuned LoRA Adapter in Figure 33. Due to the consistency in most parameters, there are no significant changes in behaviours of the Hessian spectrum at different checkpoints of fine-tuning, nor the minima sharpness.

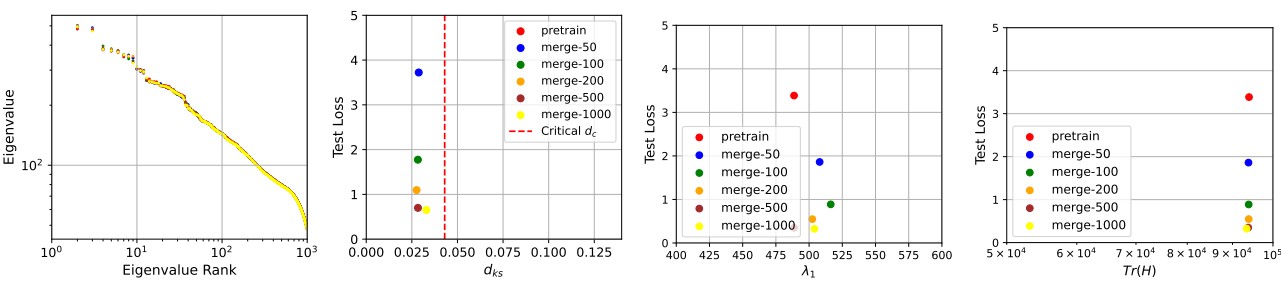

*Figure 33.* Due to consistency in most parameters, the Hessian spectrum does not change significantly at different fine-tuning checkpoints of the merged TinyLlama with LoRA adapter. The observation demonstrated the minimal impact to the original model when tuning with Low-Rank Adaptations. Model: TinyLlama-1.1B-Chat-v1.0 (merged with LoRA adapter). Dataset: `MathQA`.

## F. Kolmogorov-Smirnov Goodness-of-Fit Test

In this section, we introduce how to conduct the Kolmogorov-Smirnov Goodness-of-Fit Test for the self-containedness purpose.

As we mentioned above, our work used Maximum Likelihood Estimation (MLE) (Myung, 2003; Clauset et al., 2009) for estimating the parameter $\beta$ of the fitted power-law distributions and the Kolmogorov-Smirnov Test (KS Test) (Massey Jr, 1951; Goldstein et al., 2004) for statistically testing the goodness of the fit. The KS test statistic is the KS distance $d_{\mathrm{ks}}$ between the hypothesized (fitted) distribution and the empirical data, which measures the goodness of fit. Mathematically, the KS distance is defined as

$$d_{\mathrm{ks}} = \sup_{\lambda} |F^{\star}(\lambda) - \hat{F}(\lambda)|, \tag{15}$$

where $F^{\star}(\lambda)$ is the hypothesized cumulative distribution function and $\hat{F}(\lambda)$ is the empirical cumulative distribution function based on the sampled data (Goldstein et al., 2004). The estimated power exponent via MLE (Clauset et al., 2009) can be written as

$$\hat{\beta} = 1 + K \left[ \sum_{i=1}^{K} \ln \left( \frac{\lambda_i}{\lambda_{\mathrm{cutoff}}} \right) \right]^{-1}, \tag{16}$$

where $K$ is the number of tested samples and we set $\lambda_{\mathrm{cutoff}} = \lambda_k$. We note that the Powerlaw library (Alstott et al., 2014) provides a convenient tool to compute the KS distance, $d_{\mathrm{ks}}$, and estimate the power exponent.

According to the practice of KS Test (Massey Jr, 1951), we first state **the power-law hypothesis** that the tested spectrum is power-law. If $d_{\mathrm{ks}}$ is higher than the critical value $d_{\mathrm{c}}$ at the $\alpha = 0.05$ significance level, the KS test statistically will support the power-law hypothesis (we cannot reject the power-law hypothesis). We display the critical values in Table 6.

We conducted the KS tests for all of our studied spectra. We display the KS test statistics and the estimated power exponents $\hat{\beta}$ with standard errors $\sigma$ as well as the corresponding $\hat{s}$ in Tables 7, 8, 9, and 10. In the tables, we take the base hyperparameter setting in Appendix C as the default setting. For better visualization, we color accepting the power-law hypothesis in blue and color rejecting the power-law hypothesis (and the cause) in red.

*Table 6.* The Table of Kolmogorov-Smirnov Test Critical Values (Significance Level), which was first reported in Massey Jr (1951). If the KS distance $d_{\mathrm{ks}}$ is lower than a critical value, such as $\frac{1.36}{\sqrt{K}}$ , we would reject the null hypothesis and accept the power-law hypothesis at the $\alpha = 0.05$ significance level. Note that $K$ is the number of tested eigenvalues.

| Sample size | $\alpha = 0.2$ | $\alpha = 0.15$ | $\alpha = 0.1$ | $\alpha = 0.05$ | $\alpha = 0.01$ |
|---|---|---|---|---|---|
| K > 35 | $\frac{1.07}{\sqrt{k}}$ | $\frac{1.14}{\sqrt{k}}$ | $\frac{1.22}{\sqrt{k}}$ | $\frac{1.36}{\sqrt{k}}$ | $\frac{1.63}{\sqrt{k}}$ |
| K = 50 | 0.151 | 0.161 | 0.173 | 0.192 | 0.231 |
| K = 1000 | 0.0338 | 0.0360 | 0.0386 | 0.0430 | 0.0515 |

*Table 7.* The Kolmogorov-Smirnov statistics of LeNet on MNIST and Fashion MNIST. The estimated power exponent $\hat{\beta}$ and slope magnitude $\hat{s}$ are also displayed.

| Dataset | Model | Training | Sample size | Setting | $d_{\text{ks}}$ | $d_{\text{c}}$ | Power-Law | $\hat{\beta} \pm \sigma$ | $\hat{s}$ |
|---|---|---|---|---|---|---|---|---|---|
| MNIST | LeNet | Random | 1000 | - | 0.0796 | 0.0430 | No | | |
| MNIST | LeNet | SGD | 1000 | - | 0.00900 | 0.0430 | Yes | $1.991 \pm 0.031$ | 1.009 |
| MNIST | LeNet | Vanilla SGD | 1000 | - | 0.0103 | 0.0430 | Yes | $1.914 \pm 0.029$ | 1.094 |
| MNIST | LeNet | Adam | 1000 | - | 0.00962 | 0.0430 | Yes | $1.873 \pm 0.028$ | 1.145 |
| MNIST | LeNet | AMSGrad | 1000 | - | 0.00987 | 0.0430 | Yes | $1.845 \pm 0.027$ | 1.184 |
| MNIST | LeNet | AdaBound | 1000 | - | 0.00889 | 0.0430 | Yes | $1.904 \pm 0.029$ | 1.106 |
| MNIST | LeNet | Yogi | 1000 | - | 0.00966 | 0.0430 | Yes | $1.834 \pm 0.026$ | 1.198 |
| MNIST | LeNet | RAdam | 1000 | - | 0.0164 | 0.0430 | Yes | $1.889 \pm 0.028$ | 1.125 |
| MNIST | LeNet | Adai | 1000 | - | 0.0101 | 0.0430 | Yes | $1.892 \pm 0.028$ | 1.122 |
| MNIST | LeNet | PNM | 1000 | - | 0.0127 | 0.0430 | Yes | $1.846 \pm 0.027$ | 1.181 |
| MNIST | LeNet | Lookahead | 1000 | - | 0.0101 | 0.0430 | Yes | $1.982 \pm 0.031$ | 1.018 |
| MNIST | LeNet | DiffGrad | 1000 | - | 0.0105 | 0.0430 | Yes | $1.834 \pm 0.026$ | 1.198 |
| MNIST | LeNet | SGD | 1000 | $B = 128$ | 0.00900 | 0.0430 | Yes | $1.991 \pm 0.031$ | 1.009 |
| MNIST | LeNet | SGD | 1000 | $B = 512$ | 0.00787 | 0.0430 | Yes | $1.894 \pm 0.028$ | 1.119 |
| MNIST | LeNet | SGD | 1000 | $B = 640$ | 0.0125 | 0.0430 | Yes | $1.838 \pm 0.027$ | 1.194 |
| MNIST | LeNet | SGD | 1000 | $B = 768$ | 0.278 | 0.0430 | No | | |
| MNIST | LeNet | SGD | 1000 | $B = 1024$ | 0.129 | 0.0430 | No | | |
| MNIST | LeNet | SGD | 1000 | $B = 8192$ | 0.240 | 0.0430 | No | | |
| MNIST | LeNet | SGD | 1000 | $B = 16384$ | 0.249 | 0.0430 | No | | |
| MNIST | LeNet | SGD | 1000 | $B = 32768$ | 0.201 | 0.0430 | No | | |
| MNIST | LeNet | SGD | 1000 | $B = 50000$ | 0.139 | 0.0430 | No | | |
| MNIST | LeNet | SGD | 1000 | $B = 60000$ | 0.0936 | 0.0430 | No | | |
| MNIST | LeNet | SGD | 1000 | $N = 600$ | 0.205 | 0.0430 | No | | |
| MNIST | LeNet | SGD | 1000 | $N = 800$ | 0.0399 | 0.0430 | Yes | $1.995 \pm 0.031$ | 1.004 |
| MNIST | LeNet | SGD | 1000 | $N = 1000$ | 0.0198 | 0.0430 | Yes | $2.128 \pm 0.036$ | 0.886 |
| MNIST | LeNet | SGD | 1000 | $N = 3000$ | 0.0159 | 0.0430 | Yes | $2.091 \pm 0.034$ | 0.917 |
| MNIST | LeNet | SGD | 1000 | $N = 6000$ | 0.0151 | 0.0430 | Yes | $2.001 \pm 0.032$ | 0.999 |
| MNIST | LeNet | SGD | 1000 | 40% Label Noise | 0.180 | 0.0430 | No | | |
| MNIST | LeNet | SGD | 1000 | 80% Label Noise | 0.157 | 0.0430 | No | | |
| MNIST | LeNet | SGD | 1000 | Random Labels | 0.0482 | 0.0430 | No | | |
| Fashion-MNIST | LeNet | Random | 1000 | - | 0.0971 | 0.0430 | No | | |
| Fashion-MNIST | LeNet | SGD | 1000 | - | 0.0132 | 0.0430 | Yes | $1.939 \pm 0.030$ | 1.065 |
| MNIST | LeNet | SGD | 1000 | Eigengap | 0.0153 | 0.0430 | Yes | $1.550 \pm 0.017$ | 1.817 |
| Fashion-MNIST | LeNet | SGD | 1000 | Eigengap | 0.0240 | 0.0430 | Yes | $1.520 \pm 0.017$ | 1.922 |
| MNIST | LeNet | SGD | 1000 | Epoch= 1 | 0.0321 | 0.0430 | Yes | $1.908 \pm 0.029$ | 1.102 |
| MNIST | LeNet | SGD | 1000 | Epoch= 2 | 0.0298 | 0.0430 | Yes | $1.920 \pm 0.029$ | 1.087 |
| MNIST | LeNet | SGD | 1000 | Epoch= 3 | 0.0354 | 0.0430 | Yes | $1.916 \pm 0.031$ | 1.092 |
| MNIST | LeNet | SGD | 1000 | Epoch= 10 | 0.0291 | 0.0430 | Yes | $2.029 \pm 0.033$ | 0.972 |
| MNIST | LeNet | SGD | 1000 | Epoch= 20 | 0.0268 | 0.0430 | Yes | $2.081 \pm 0.034$ | 0.925 |
| MNIST | LeNet | SGD | 1000 | Epoch= 30 | 0.0068 | 0.0430 | Yes | $2.074 \pm 0.034$ | 0.930 |

*Table 8.* The Kolmogorov-Smirnov statistics of LeNet on CIFAR-10 and CIFAR-100. The estimated power exponent $\hat{\beta}$ and slope magnitude $\hat{s}$ are also displayed.

| Dataset | Model | Training | Sample size | Setting | $d_{\mathrm{ks}}$ | $d_{\mathrm{c}}$ | Power-Law | $\hat{\beta} \pm \sigma$ | $\hat{s}$ |
|---------|-------|----------|-------------|---------|-------------------|------------------|-----------|--------------------------|-----------|
| CIFAR-10 | LeNet | Random | 1000 | - | 0.0663 | 0.0430 | No | | |
| CIFAR-10 | LeNet | SGD | 1000 | - | 0.0279 | 0.0430 | Yes | $1.968 \pm 0.031$ | 1.033 |
| CIFAR-10 | LeNet | Vanilla SGD | 1000 | - | 0.0276 | 0.0430 | Yes | $1.935 \pm 0.030$ | 1.069 |
| CIFAR-10 | LeNet | Adam | 1000 | - | 0.0269 | 0.0430 | Yes | $1.806 \pm 0.025$ | 1.241 |
| CIFAR-10 | LeNet | AMSGrad | 1000 | - | 0.0232 | 0.0430 | Yes | $1.786 \pm 0.025$ | 1.271 |
| CIFAR-10 | LeNet | AdaBound | 1000 | - | 0.0297 | 0.0430 | Yes | $1.901 \pm 0.028$ | 1.110 |
| CIFAR-10 | LeNet | Yogi | 1000 | - | 0.0184 | 0.0430 | Yes | $1.806 \pm 0.025$ | 1.241 |
| CIFAR-10 | LeNet | RAdam | 1000 | - | 0.0163 | 0.0430 | Yes | $1.733 \pm 0.023$ | 1.363 |
| CIFAR-10 | LeNet | Adai | 1000 | - | 0.0310 | 0.0430 | Yes | $1.918 \pm 0.029$ | 1.090 |
| CIFAR-10 | LeNet | PNM | 1000 | - | 0.0347 | 0.0430 | Yes | $1.911 \pm 0.029$ | 1.098 |
| CIFAR-10 | LeNet | Lookahead | 1000 | - | 0.0358 | 0.0430 | Yes | $1.964 \pm 0.030$ | 1.037 |
| CIFAR-10 | LeNet | DiffGrad | 1000 | - | 0.0303 | 0.0430 | Yes | $1.803 \pm 0.024$ | 1.236 |
| CIFAR-100 | LeNet | Random | 1000 | - | 0.0944 | 0.0430 | No | | |
| CIFAR-100 | LeNet | SGD | 1000 | - | 0.0315 | 0.0430 | Yes | $1.908 \pm 0.029$ | 1.101 |
| CIFAR-100 | LeNet | Vanilla SGD | 1000 | - | 0.0379 | 0.0430 | Yes | $1.903 \pm 0.029$ | 1.108 |
| CIFAR-100 | LeNet | SGD | 1000 | Evaluated on CIFAR-10 | 0.0306 | 0.0430 | Yes | $1.913 \pm 0.029$ | 1.095 |

*Table 9.* The Kolmogorov-Smirnov statistics of FCN. The estimated power exponent $\hat{\beta}$ and slope magnitude $\hat{s}$ are also displayed.

| Dataset | Model | Training | Sample size | Setting | $d_{\mathrm{ks}}$ | $d_{\mathrm{c}}$ | Power-Law | $\hat{\beta} \pm \sigma$ | $\hat{s}$ |
|---------|-------|----------|-------------|---------|-------------------|------------------|-----------|--------------------------|-----------|
| Avila | 2Layer-FCN | SGD | 50 | - | 0.0683 | 0.176 | Yes | $1.604 \pm 0.085$ | 1.656 |
| MNIST | 1Layer-FCN | Random | 1000 | - | 0.185 | 0.0430 | No | | |
| MNIST | 1Layer-FCN | SGD | 1000 | - | 0.241 | 0.0430 | No | | |
| MNIST | 2Layer-FCN | Random | 1000 | - | 0.129 | 0.0430 | No | | |
| MNIST | 2Layer-FCN | SGD | 1000 | - | 0.0112 | 0.0430 | Yes | $2.209 \pm 0.038$ | 0.827 |
| MNIST | 4Layer-FCN | Random | 1000 | - | 0.0628 | 0.0430 | No | | |
| MNIST | 4Layer-FCN | SGD | 1000 | - | 0.0141 | 0.0430 | Yes | $2.201 \pm 0.038$ | 0.833 |
| MNIST | 2Layer-FCN | SGD | 1000 | Width=10 | 0.149 | 0.0430 | No | | |
| MNIST | 2Layer-FCN | SGD | 1000 | Width=20 | 0.185 | 0.0430 | No | | |
| MNIST | 2Layer-FCN | SGD | 1000 | Width=30 | 0.0656 | 0.0430 | No | | |
| MNIST | 2Layer-FCN | SGD | 1000 | Width=50 | 0.0187 | 0.0430 | Yes | $2.138 \pm 0.028$ | 0.879 |
| MNIST | 2Layer-FCN | SGD | 1000 | Width=70 | 0.0376 | 0.0430 | Yes | $2.271 \pm 0.030$ | 0.787 |
| MNIST | 2Layer-FCN | SGD | 1000 | Width=100 | 0.0112 | 0.0430 | Yes | $2.209 \pm 0.038$ | 0.827 |

*Table 10.* The Kolmogorov-Smirnov statistics of ResNet18. The estimated power exponent $\hat{\beta}$ and slope magnitude $\hat{s}$ are also displayed.

| Dataset | Model | Training | Sample size | Setting | $d_{\mathrm{ks}}$ | $d_{\mathrm{c}}$ | Power-Law | $\hat{\beta} \pm \sigma$ | $\hat{s}$ |
|---------|-------|----------|-------------|---------|--------|------|-----------|-----------------|------|
| CIFAR-10 | ResNet18 | Random | 50 | - | 0.334 | 0.176 | No | | |
| CIFAR-10 | ResNet18 | SGD | 50 | - | 0.0803 | 0.176 | Yes | $2.146 \pm 0.162$ | 0.873 |
| CIFAR-10 | ResNet18 | Vanilla SGD | 50 | - | 0.0891 | 0.176 | Yes | $2.193 \pm 0.169$ | 0.838 |
| CIFAR-10 | ResNet18 | Adam | 50 | - | 0.0478 | 0.176 | Yes | $2.062 \pm 0.149$ | 0.950 |
| CIFAR-10 | ResNet18 | AMSGrad | 50 | - | 0.0542 | 0.176 | Yes | $2.041 \pm 0.147$ | 0.961 |
| CIFAR-10 | ResNet18 | AdaBound | 50 | - | 0.0588 | 0.176 | Yes | $2.029 \pm 0.146$ | 0.971 |
| CIFAR-10 | ResNet18 | Yogi | 50 | - | 0.116 | 0.176 | Yes | $1.915 \pm 0.129$ | 1.092 |
| CIFAR-10 | ResNet18 | RAdam | 50 | - | 0.168 | 0.176 | Yes | $1.794 \pm 0.1112$ | 1.259 |
| CIFAR-10 | ResNet18 | Adai | 50 | - | 0.103 | 0.176 | Yes | $2.183 \pm 0.167$ | 0.845 |
| CIFAR-10 | ResNet18 | PNM | 50 | - | 0.138 | 0.176 | Yes | $2.132 \pm 0.160$ | 0.884 |
| CIFAR-10 | ResNet18 | Lookahead | 50 | - | 0.110 | 0.176 | Yes | $2.098 \pm 0.155$ | 0.911 |
| CIFAR-10 | ResNet18 | DiffGrad | 50 | - | 0.068 | 0.176 | Yes | $2.055 \pm 0.149$ | 0.948 |
| CIFAR-10 | ResNet18 | SGD | 50 | $B = 512$ | 0.0561 | 0.176 | Yes | $2.146 \pm 0.151$ | 0.936 |
| CIFAR-10 | ResNet18 | SGD | 50 | $B = 1024$ | 0.0647 | 0.176 | Yes | $2.076 \pm 0.152$ | 0.929 |
| CIFAR-10 | ResNet18 | SGD | 50 | $B = 1152$ | 0.0598 | 0.176 | Yes | $2.060 \pm 0.150$ | 0.944 |
| CIFAR-10 | ResNet18 | SGD | 50 | $B = 1280$ | 0.331 | 0.176 | No | | |
| CIFAR-10 | ResNet18 | SGD | 50 | $B = 2048$ | 0.334 | 0.176 | No | | |
| CIFAR-10 | ResNet18 | SGD | 50 | $B = 4096$ | 0.334 | 0.176 | No | | |
| CIFAR-10 | ResNet18 | SGD | 50 | $B = 16384$ | 0.343 | 0.176 | No | | |
| CIFAR-100 | ResNet18 | Random | 50 | - | 0.373 | 0.176 | No | | |
| CIFAR-100 | ResNet18 | SGD | 50 | - | 0.108 | 0.176 | Yes | $2.299 \pm 0.184$ | 0.770 |

