# OpenReview forum: "Investigating the Overlooked Hessian Structure: From CNNs to LLMs"
_ICML.cc/2025/Conference — ICML 2025 poster_

### Official Review · Reviewer_qcZb · 2025-03-06

**Overall Recommendation:** 3

**Summary:**

In this work, the authors report a previously overlooked power-law Hessian structure in well-trained deep neural networks, which includes CNNs and LLMs. The authors show that both the top few thousand eigenvalues as well as the eigengaps follow approximately a power-law $p(\lambda) = Z_c^{-1} \lambda^{-\beta}$, which they test statistically using the Kolmogorov-Smirnov Test (KS Test). This phenomenon is observed across multiple optimizers, datasets, and architectures and differs from the eigenspectrum of randomly initialed networks.
In addition, the authors provide a theoretical interpretation via the maximum-entropy principle.
The authors also provide additional ablations, including different optimizers, overparametrizing the network, the size of training data, and the batch size, and show how the fitted slope magnitude correlates well with the largest Hessian eigenvalue, the Hessian trace, and the test performance and how the KS distance $d_{KS}$ predicts the generalization performance of LLMs models, which conventional sharpness measures ($Tr(H)$ and $\lambda_{\max}$) fail to capture.

**Claims And Evidence:**

Generally, I think that all claims are supported by clear and convincing evidence. For some of the experiments, I believe that one needs to extend the figures to fully make the claims of the authors. For instance, in the experiment on varying Batch Size, where the authors describe three different phases of large-batch training, it would be beneficial to also include the Test Error to Figure 8 to more clearly different phases apart from the qualitative shape of the eigenvalue distribution. Actually, it would be interesting to see if one could have a similar plot as e.g. Fig. 10 or Fig. 11, where the $d_{KS}$ for each batch size is plotted against the Test Loss/Error.

**Essential References Not Discussed:**

-

**Experimental Designs Or Analyses:**

The experimental designs and resulting analyses are sound as I have already elaborated in the section above on "Claims and Evidence".
As elaborated above, it would be nice to provide further figures in the setting of image classification, similar to the section on LLMs. e.g. plotting $d_{KS}$ against the test loss for the ablation study on large-batch training and training-data size.

**Methods And Evaluation Criteria:**

The methods and evaluation criteria seem reasonable to me. The authors chose a reasonable range of different models, including FCN, LeNet, ResNet18, and ViTs for some image classification datasets, including MNIST, Fashion-MNIST, Cifar-10/100, and non-image Avila as well as GPT2-models on OpenWebText, Shakespeare, and MathQA.
Also, the ablation studies were quite thorough, including training data size, model size, batch size, and optimizers.
Of course, it would be beneficial if the results could also be reproduced in some other settings, as Figure 3, Figure 4 and Figure 7 are all conducted on LeNet trained on MNIST. Is it possible to reproduce some of the results for instance on ResNet18 trained on Cifar10?

**Other Comments Or Suggestions:**

- some minor typos: line 258: considered, line 389: maximum, line 429: considered

- it was not very clear to me what the three phases of large-batch training were until I read the section in the appendix. I would recommend the authors either to refer to this section in the main paper or to move this section from the appendix to the main paper to avoid confusion.

**Other Strengths And Weaknesses:**

I think the observation of the power-law distribution of the eigenvalues that the authors report is quite intriguing and it's exciting to see it occur in such various settings. It is also interesting to see how the slope magnitude correlates with other sharpness measures and $d_{KS}$ correlates well with generalization in LLMs.

I believe that this is very interesting work and am open to raising my score if the authors address my comments and questions.

**Questions For Authors:**

- What is meant by the so-called equilibrium of DNNs, which is mentioned in line 124? Is it equivalent to a DNN at convergence?

- I am not entirely sure, but the eigenvalue distribution of sharp minima (e.g. large-batch training with B=768) intuitively seems to obey the power-law distribution less than a randomly initialized network, which to me implies a larger $d_{KS}$. However, I would assume that the network trained with $B=768$ will still perform better than random. (This is also why I asked for additional plots.) Can the authors explain this? Or for which cases $d_{KS}$ is a good predictor of generalization?

- I checked the work by Wu et al. [2017] and noticed therein in Figure 2 (right) that models with quite different generalization behavior (21% vs. 92.9%) have a similar shape of the spectrum (at least by eyeballing), but that the spectrum is simply shifted. Can the authors elaborate how this result fits into the analysis that the authors have conducted? (e.g. Figures 3 and 4 in this work?)

**Relation To Broader Scientific Literature:**

The analysis of $d_{KS}$ as a generalization measure in LLMs relates to previous results by Jiang et al. [2019] and Kaur et al. [2023], which uses sharpness measures such as the Hessian trace $\text{Tr}(H)$ and the largest eigenvalue $\lambda_{\max}(H)$ as a measure of generalization.

The power-law Hessian structure relates to previous work analyzing the Hessian structure of DNNs, including work by Pennington & Bahri [2017], who study the eigenvalue distribution through random matrix theory; Singh et al. [2021], who study the rank of DNNs; Liao & Mahoney [2021], who study the Hessian spectra of nonlinear models and Dauphin et al. [2024], who investigate the Nonlinear Modeling Error (NME) matrix part of the Hessian, which has been neglected in previous analysis.

**Theoretical Claims:**

I only checked the proof for Theorem 1 in Appendix A and think that it is fine as is.

---

> ### Author Rebuttal · Authors · 2025-04-01
>
> We appreciate your insightful feedback. Below, we duly address your concerns through additional experiments and careful responses.
>
> Q1: Is it possible to reproduce some of the results for instance on ResNet18 trained on Cifar10?
>
> A1. Yes, it is possible. The Kolmogorov-Smirnov distance decreases from d_KS = 0.198 in randomly initialized ResNet18 to d_KS = 0.031 (< d_c) in the well-trained model, providing further empirical support for our paper's central conclusion. We expect to supply these results in the revision.
>
> Q2: The experimental designs and resulting analyses are sound, as it would be nice to provide further figures in the setting of image classification, similar to the section on LLMs.
>
> A2. Thank you for your complementation and suggestion. We will supply these results in the revision.
>
> Q3: It is also interesting to see how the slope magnitude correlates with other sharpness measures and correlates well with generalization in LLMs.
>
> A3. Thank you for your suggestion. However, we are unable to identify significant correlations between slope and other metrics on LLMs while it works well on CNNs. In particular, for GPT2-small fine-tuned on the Shakespeare dataset, the Pearson correlation coefficient of the slope magnitude to test loss and hessian trace are -0.173 and 0.414, drawing a negative conclusion on the validity of using the Hessian spectrum slope as a generalization metric.
>
> Q4: What is meant by the so-called equilibrium of DNNs, which is mentioned in line 124?
>
> A4: Thank you for raising this question. We have followed the concept from statistical physics (e.g. Boltzmann Machine) we referenced in line 117. Equilibrium often indicates that a system or a network reaches a stationary distribution, which can help in understanding the optimization dynamics of NNs in some ML works. However, we would like to remove the contents of equilibrium, as it is not a necessary concept in our work.
>
> Q5: Can the authors explain the eigenvalue distribution of sharp minima (e.g. large-batch training with B=768) intuitively seems to obey the power-law distribution less than a randomly initialized network?
>
> A5: Thank you for your question. We have discussed the observation that large-batch trained networks (B > 768) do not exhibit power laws, resembling under-parameterized networks in Appendix D, and summarized as the three phases in large-batch training. Our Hessian spectra analysis discovery differs from traditional beliefs that different phases exist in training. In phase 2, the Hessian eigenvalue increases and breaks the power-law structure, while the model's performance is much superior compared to untrained neural networks. We may leave a deeper investigation for future works.
>
> Q6: Can the authors elaborate how the work by Wu et al. [2017] (Figure 2 right) fits into the analysis?
>
> A6: Thank you for your suggestion. Wu et al. utilize an attack set to control the model's performance which differs from the principle experiment setup in our main paper. We believe the behaviour cannot be guaranteed to be consistent across different data recipes and setups. We have instead discussed a similar setting with noisy labels presented in Figure 15, Appendix D. The red line refers to the model with the best generalization performance as trained on the fully clean dataset, corresponds to the red line of the model with 92.92% accuracy in Figure 2 (right) of Wu et al. (2017). We may observe that the power-law behaviour of the Hessian spectrum computed on the Test set is disturbed by the noisy labels introduced. We may also observe that the magnitude of eigenvalues increases for networks trained with adversarial labels and poorer generalization performance.
>
> Finally, we sincerely thank the reviewer's feedback. It definitely inspires us to further improve our work and clarification for more readers. We respectfully hope that the reviewer can reevaluate our work given the responses addressing your main concerns.

---

> > ### Comment · Reviewer_qcZb · 2025-04-03
> >
> > I would like to thank the authors for their rebuttal.
> >
> > A1. Thank you for the additional experiments. Have you tried using multiple optimizers to get a similar picture as Figure 3 and 4? Can you also provide the figures via an anonymized link?
> >
> > A3. Thank you for the answer. So it seems like the $d_{KS}$ measure is more predictive in this case. I just compared the results in Figure 11 on fine-tuning tasks with the results in Figure 31 on LoRA-Adapters, where the picture seems reverted and the Hessian trace correlates very well with generalization, while $d_{KS}$ does not. Do the authors also have a possible explanation in what ways full-parameter fine-tuning differs from parameter-efficient fine-tuning tasks such as with LoRA?
> >
> > Q5./Q6. I was raising these two questions because it would put the $d_{KS}$ measure as a predictor for generalization into question, if there are certain choices in training (e.g. large batch training) which lead to a well performing solution, while not having the power-law structure of the Hessian.
> >
> > Despite this intriguing observation on the power-law structure in well trained networks (in most cases), it remains unclear to me when $d_{KS}$ is a good measure and in which cases the slope of the power-law is also insightful. (A3)

---

> > > ### Author Response · Authors · 2025-04-09
> > >
> > > **Q7 (to A1):** Can you also provide the figures using multiple optimizers as Figure 3 and 4 via an anonymized link?
> > >
> > > **A7:** Thank you for your suggestions. Sadly, due to the constrained time limit in this rebuttal round, we cannot provide the full experiment results on multiple optimizers. We have provided the experiment results for Optimizer SGD and Adam in the link below: \url{https://anonymous.4open.science/r/anonymous_link_11101-5D5D/README.md}. The experiment setups followed the CIFAR-10 settings in Appendix C.1.3. We wish to supply further results in the future.
> > >
> > > **Q8 (to A3):** Do the authors also have a possible explanation in what ways full-parameter fine-tuning differs from parameter-efficient fine-tuning tasks such as with LoRA? + In which cases the slope of the power-law is also insightful?
> > >
> > > **A8:**  Regarding Figures 11 and 31, we note that no existing literature indicates that the Hessian of low-rank adapters accurately captures the true loss landscape of the original model or reliably reflects the flatness or sharpness of its minima.  In pursuit of an intuitive explanation, the observed pattern of an initial increase followed by a decrease may indicate a transition from one local minimum to another within the loss landscape. We further hypothesize that the power-law slope only becomes informative when the power-law behaviour is highly consistent, as indicated by a significantly small d_KS; in such cases, the slope is more likely to have a stronger impact on the model's performance.
> > >
> > > **Q9 (to Q5/6 & A3):** Despite this intriguing observation on the power-law structure in well-trained networks (in most cases), it remains unclear when d_KS is a good measure.
> > >
> > > **A9:**  Thank you for this insightful question. In our paper, we revealed the Hessian power-law structure and demonstrated its widespread existence in deep learning models. This power-law structure itself is a novel and promising perspective on understanding DNNs. Building on this power-law hypothesis and our empirical observations, we introduced the KS distance as a novel measure and explored its connection to the loss landscape, the sharpness of minima, and generalization behaviour — offering what we believe is a valuable contribution to the field. Furthermore, while existing literature highlights sharpness-based metrics as promising directions [1, 2], they often fall short for LLMs, whereas the KS distance tends to correlate more consistently with generalization in many cases. We recognize the importance of further investigating the conditions under which d_KS serves as a reliable metric, which we plan to pursue through more extensive experimentation across diverse settings. Nevertheless, we acknowledge that no single generalization metric can universally apply across all deep learning settings—considering the wide variability in model architectures, datasets, tasks, batch sizes, and optimizers [3].  We hope that our work, alongside other studies exploring the strengths and limitations of various generalization metrics, contributes to a deeper understanding of deep neural networks and their behaviour under different conditions.
> > >
> > > Refs.
> > >
> > > [1] Jiang, Yiding, et al. "Fantastic Generalization Measures and Where to Find Them." ICLR 2020.
> > >
> > > [2] Dziugaite, Gintare Karolina, et al. "In search of robust measures of generalization." Advances in Neural Information Processing Systems 33 (2020): 11723-11733.
> > >
> > > [3] Gastpar, Michael, et al. "Fantastic Generalization Measures are Nowhere to be Found." ICLR 2024.

---

### Official Review · Reviewer_sJoB · 2025-03-14

**Overall Recommendation:** 3

**Summary:**

This paper investigates the power-law structure of the Hessian matrix in deep neural networks, including Convolutional Neural Networks (CNNs) and Large Language Models (LLMs). Key contributions include: A maximum-entropy principle from statistical physics is proposed to explain the emergence of the power-law structure, linking it to flat minima and generalization. The authors identify a power-law distribution in the Hessian spectra of well-trained NNs, contrasting with random or under-trained models. This structure is consistent across CNNs, LLMs, and Vision Transformers (ViTs). This paper indicate this power-law structure helps predict the generalization of LLMs.

**Claims And Evidence:**

The claims in the submission are generally supported by a theoretical interpretation and experimental discovery. However, the theoretical analysis here is merely an intuitive explanation and does not provide a rigorous proof.

**Essential References Not Discussed:**

No.

**Ethical Review Concerns:**

No ethical concerns identified. This work focuses on the empirical performance of existing networks.

**Experimental Designs Or Analyses:**

From an experimental perspective, the experimental design of this paper can illustrate the claims.

**Methods And Evaluation Criteria:**

The evaluation criteria for generalization are widely recognized and reasonable standards.

**Other Comments Or Suggestions:**

See the following Questions.

**Other Strengths And Weaknesses:**

The theoretical analysis is the weakness of this article; the author only somewhat tenuously accepts the maximum entropy principle and can not
provide a rigorous explanation of the relationship between the two sides. However, this article still provides a new perspective on how to predict
the generalization of LMMs.

**Questions For Authors:**

1. The author should mention the computational cost and time required to obtain the Hessian matrix and its spectrum. This could become unaffordable when applied to advanced LMMs?

2. In the evaluation criteria of this article, when the KS distance exceeds 0.05, it is considered as not satisfying the power-law hypothesis. However, as shown in Table 1, the DS distance of untrained networks is also quite small. Is this value a widely recognized criteria, or could it be that many untrained models themselves also satisfy the power-law hypothesis?

**Relation To Broader Scientific Literature:**

This article adopts the well-known maximum entropy principle to intuitively explain the power-law distribution of the eigenvalues of the network Hessian matrix. The main contribution lies in extensive experimental observations, which is not very related to prior works.

**Theoretical Claims:**

The proof of the theory are correct but somehow straightforward.

---

> ### Author Rebuttal · Authors · 2025-04-01
>
> Thank you for your support of our work and constructive suggestions. We try our best to structure your concerns and duly address them as follows:
>
> Q1: The author cannot provide a rigorous explanation of the maximum entropy principle. However, this article still provides a new perspective on how to predict the generalization of LMMs.
>
> A1: Thanks for the suggestion. First, to recognize our works, we would like to convey that we are the first to provide an interpretation of the power-law hypothesis while following the maximum entropy principle in statistical physics. We refer you to the works (Bahri et al., 2020; Torlai & Melko, 2016) we referenced in line 117 for better understanding. To ground our approach, we draw parallels with research works in the machine learning field that adopt similar theoretical frameworks: Dinh et al. studied minima flatness with similar Hessian-based measures [1] and Baldassi et al. also interpreted the minima flatness of neural networks from an entropy perspective [2]. However, we understand that this work lacks a strong justification for using the maximum entropy principle for power-law interpretation. We are willing to improve the theoretical analysis and justify the relationship between the maximum entropy principle and minima flatness of neural networks in future revisions.
>
> Q2: The author should mention the computational cost and time required to obtain the Hessian matrix and its spectrum.
>
> A2: Thanks for your suggestion. We employed the Stochastic Lanczos Quadrature (SLQ) algorithm with optimizations tailored for large neural networks in our work. SLQ consists of K Lanczos iterations and M integral steps to reduce estimation error. The primary computational expense arises from repeated Hessian-vector product (HvP) computations in each Lanczos step, requiring two backpropagation passes, which we implemented using PyTorch's auto-differentiation framework. To manage memory overhead, we stored intermediate SLQ computations as temporary tensors on disk, reducing space requirements to a level comparable to training the network via SGD with momentum.
>
> Assuming the time complexity of a single backpropagation step is O(N), the overall SLQ complexity is O(N^2 \cdot K \cdot M). For all LLM experiments presented in our paper, we set M = 3, K = 1000 and a sampled batch size of 256 as in Zhang et al. (2024b). In our implementation, computing 3000 Hessian eigenvalues for the last layer of GPT2-small takes around 12-14 GPU hours on a standard RTX 4090. While PyTorch's current implementation does not support explicit storage of the computation graph, in theory, the time complexity could be further reduced to O(N \cdot K \cdot M) by pre-saving gradients and the computation graph before SLQ execution. Consequently, computing the top-N eigenvalues has a time complexity equivalent to training the network for N steps.
>
> Q3: Is the small KS distance of untrained networks widely recognized criteria, or could it be that many untrained models themselves also satisfy the power-law hypothesis?
>
> A3: We do not believe that many untrained models could satisfy the power-law hypothesis. We would like to convey that the critical values (d_C) of the KS Tests are derived from a=0.05 significance level (Massey Jr, 1951) and we would reject the null hypothesis for d_KS below the critical values. As we have conducted a large number of experiments on models from CNNs to LLMs, we have not observed the d_KS of any untrained networks are below d_C. Some untrained models may have a close d_KS but they are still far from accepting the power-law hypothesis.
>
> Finally, we sincerely thank the reviewer's feedback. It definitely inspires us to further improve our work and clarification for more readers. We would highly appreciate it if you could kindly reconsider the rating of our work given the initial score of 3 and the addressed concerns. Your suggestion and action will both make an addition to our community.
>
> Refs:
>
> [1] Dinh, Laurent, et al. "Sharp minima can generalize for deep nets." ICML, 2017.
>
> [2] Baldassi, Carlo, Fabrizio Pittorino, and Riccardo Zecchina. "Shaping the learning landscape in neural networks around wide flat minima." PNAS, 2020.

---

### Official Review · Reviewer_tKCL · 2025-03-14

**Overall Recommendation:** 2

**Summary:**

This paper investigates the Hessian structure in CNNs and LLMs from the point of view of the power laws in the Hessian spectrum. It is shown that across a range of settings the power law like trend holds for the Hessian eigenvalues of a trained network, which however is not the case at random initialization. There is some rough theoretical interpretation that is provided based on the maximum-entropy principle. In addition to power laws in the Hessian spectrum, similar thing is shown for eigengaps. Alongside, there is a thorough empirical exploration of the effects of power law in regards to training dataset size, batch size, and some interesting trend is shown for its ability to correlate with the generalization during the course of pre-training/fine-tuning.

**Claims And Evidence:**

- In Eqn 2, when discussing the finite-sample power law, normalization is carried out by the Hessian trace. However, this may not a be suitable normalization when the Hessian is indefinite, which is definitely the case at a random initialization. A more appropriate choice would have been to consider the absolute value of the eigenvalue and then see when they are ranked, if the power law trend is present or not. Currently, I'm concerned how this would change the trend for the spectrum at random initialization.

- The link to maximum entropy is interesting, but conceiving of measure flatness of the minimum with the determinant of **Hessian inverse** is ad-hoc. Normally, you would use a spectral quantity based on the Hessian, so det(H) would be fine. I believe this is soley being done for the purposes of their interpretation, else the power law has positive exponents, which would be a mismatch. Can the authors comment on this?

**Essential References Not Discussed:**

I think the work of Mahoney and co-authors on heavy tails could be discussed and compared to, since essentially both these works are effectively comparing the trends of spectral decay.

**Experimental Designs Or Analyses:**

- The connection to generalization during training is interesting, but could use some measurements to see if the correlation is really robust or not.

- When showing the trend with different batch sizes, it is unclear what the underlying setting is. Is the number of training steps kept the same across batch sizes? How is the learning rate adjusted? These things could make a big difference

- The statement that "the model trained with limited training data finds minima with many sharp directions like under parameterized models" and their finding therein is a bit strange. Since, underparametrized is relative to number of parameters (which I believe are fixed), so actually when training set is 600, it is the most overparameterized setting. So I don't quite understand this statement in the paper and the trend.

Also, see claims and evidence section

**Methods And Evaluation Criteria:**

- I am curious if you were to look at all the eigenvalues, and not just the top 6000, how much of this behaviour holds. Of course, I understand the difficulty with such an endeavour, but this can still be done with a network having parameters in the range of 20K and see what happens.

**Other Comments Or Suggestions:**

- Table 1 caption: no \beta or slopes are mentioned

- It would be worth elaborating on the link between Eqns 1, 2, 3 to reach a wider readership.

- The 'random' in the column label 'Training' is confusing at first sight, and rather say something like, 'random-init'; else it could be thought of something else.

**Other Strengths And Weaknesses:**

The paper contains some very interesting results, on the power-law structure. However, often a lot of the results are thrown at the reader, and the overall narrative is not as cohesive. Some of the theoretical discussion is not well justified. The potential link to generalization offers an interesting avenue, perhaps that could be elaborated and studied further, to result in a more tightly-knit paper. Also, I'd move some of the other marginal analyses to the appendix.

**Questions For Authors:**

- Looking at table 1, it seems that the more complex the setting, the higher is the value of d_KS. So it seems like the power law test might fail if you go to a setting which is more complicated. Could the authors comment on the possible reason for this? Also, it would be nice to see the results on ImageNet and see what the trend is like!

- How many datapoints are used to estimate the Hessian in each of these cases in Table 1? Ideally, the power law trend should not be too sensitive to this choice of datapoints, assuming a sufficient number of samples is used. Can the authors verify this aspect?

- What do you mean by the networks being close to equilibrium, just above section 2.3? It is very vague at the moment.

- What are the numbers for GPT-2 medium and large on OpenWebText?

- It would have been interesting to read off some trends from looking at the use of different optimizer. But right now it's unclear if something can be said, except that this power law occurs for various optimizer choices.

**Relation To Broader Scientific Literature:**

This is a complementary perspective to the study of Hessian in deep learning, and the initial results like the correlation trend with generalization seem promising.

**Theoretical Claims:**

yeah theorem 1, and I think the use of Hessian inverse for determinant is a bit ad-hoc as detailed elsewhere

---

> ### Author Rebuttal · Authors · 2025-04-01
>
> Thanks for your support of our work. We try best to structure your concerns and address them as follows:
>
> Q1: Normalization by the Hessian trace in Eqn.2 may not be suitable when Hessian is indefinite.
>
> A1: Eqn.2 attempts to describe the power-law from a frequency perspective, and we reported true eigenvalues in experiments following Eqn.3 instead of being trace-normalized. Further KS tests on the absolute eigenvalues also suggest robust power-law structure.
>
> Q2: Measuring minima flatness with the Hessian inverse determinant is ad-hoc.
>
> A2: Mathematically, the absolute value of the determinant quantifies how the transformation scales n-dimensional volume, which naturally make Hessian a measure of minima flatness. It corresponds to the volume scaling factor of parallelepipeds in 3D, which implies sharp or flat curvature. We also draw parallels with research works that the Hessian determinant is a kind of sharpness measure [1] and Empirical Fisher determinant is an approximation of loss curvature [2]. We will improve the clarification in the revision.
>
> Q3: I am curious how this behavior holds at all eigenvalues.
>
> A3: Experiments on LeNet show the eigenvalues quickly decay to near zero beyond certain threshold. Researchers commonly focus on the top eigenvalues since they are significant and critically relate to the network's performance compared to smaller ones.
>
> Q4: Could use some measurements to see if the connection to generalization during training is robust / What are the numbers for GPT2 medium/large?
>
> A4: In addition to Pearson's correlation (p_corr), we have re-evaluated with Spearman's Rank Correlation (s_corr) and OLS's statistical significance (p_ols<0.05). For GPT2-{nano,small} finetuned on Shakespeare (Figure 10-11), s_corr={0.886,1.0} and p_ols={0.009,0.0}; For GPT2-{medium,large} pretrained on OpenWebText, p_corr={0.766,0.803}, s_corr={0.855,0.786} and p_ols={0.01,0.03}; the trend in the emergence of the Hessian power-law structure over training seems robust.
>
> Q5: When showing the trend with different batch sizes, the underlying setting is unclear.
>
> A5: We set epochs=50 with hyperparameters stated in Appendix C.1.2 in Table 4, thus all models are trained with the same total images. We follow your advice and use fixed total training steps, early experiments suggest that with more computation/same iterations, dKS for large-batch indeed decreases, while it is still significantly higher than small-batch training.
>
> Q6: The statement `model trained with limited training data finds minima ... like underparameterized models' is a bit strange.
>
> A6: We mean to convey that both data and parameter scaling could cause the power-law to emerge, and both scenarios of limited data and underparameterization would similarly break the power law and lead to many sharp directions in the loss landscape. We will clarify this more clearly in the revision.
>
> Q7: It seems like the power law test might fail if setting is more complicated in Table 1.
>
> A7: Thanks for this interesting question. We actually agree with your point. Too simple neural networks like LeNet may not well learn complex tasks like ImageNet, thus training loss will be very high and the power-law structure may break due to the poorly learned representation. Training LeNet however depend on ImageNet-size problems and we may supply it in the revision.
>
> Q8: How many data points are used to estimate the Hessian in Table 1?
>
> A8: All Table 1 experiments utilize the full training dataset of MNIST, CIFAR-10 etc. The same experiment on 1/10 dataset (5000 data points) has no noticeable difference.
>
> Q9: What do you mean by the networks being close to equilibrium above section 2.3?
>
> A9: Equilibrium often indicates that a system/network reaches a stationary distribution, which can help interpret NN optimization dynamics in theoretical ML works. We would remove the contents as it is not necessary in our work.
>
> Q10: It would have been interesting to read off some trends in the use of different optimizers.
>
> A10: Our findings primarily demonstrate the power-law existence, and the slope correlates to generalization and minima sharpness learned by different optimizers. A more fine relation to optimizer settings is an open question and beyond the main scope of this work.
>
> In addition to addressing the primary question, we have incorporated the reviewers’ suggestions to correct typo errors, clarify ambiguities in Eqn 1-3, change 'random' labels to 'random-init' in Figures (9,10,14), and discuss the work of Mahoney on heavy tails. We sincerely thank the reviewer's feedback and constructive suggestions. It definitely inspires us to further improve our work.
>
> We would highly appreciate it if you could kindly reconsider the rating, given the enhanced work quality under your kind help.
>
> Refs:
>
> [1] A Diffusion Theory For Deep Learning Dynamics: Stochastic Gradient Descent Exponentially Favors Flat Minima. ICLR 2021.
>
> [2] Where is the Information in a Deep Neural Network? arxiv:2019

---

### Decision · Program_Chairs · 2025-05-01

**Decision:**

Accept (poster)

**Comment:**

The paper reports a previously unnoticed power-law structure in the Hessian top eigenvalues of trained neural networks that is not present in randomly initialized nets.
It provides a theoretical motivation for this structure in terms of the maximum entropy principle.
The main contribution is an empirical study of the emergence and properties of the power law relative to relative to overparameterization and generalization on CNNs and LLMs.

This paper is a borderline case:
- The reviewers found the theoretical interpretation provided by the paper interesting but rather vague, and the authors could not substantially strengthen the intuition in the rebuttal.
- They found the empirical results presented by this work, specifically using the power law's exponent to predict generalization, to be intriguing and promising, and the experimental design to be solid.
  Some reviewers found the presentation of results non-cohesive due to the sheer amount of experiments presented in the main text.
  After reading the paper, I agree with this point.

My overall impression is that the paper contains findings to consider acceptance, but that their presentation and the story line could be improved significantly to make the main findings clearer.